



**A comparative assessment of the uncertainties of global surface-ocean $CO_2$ estimates using a**
**machine learning ensemble (CSIR-ML6 version 2019a) – have we hit the wall?**
Luke Gregor[1,2], Alice D. Lebehot[1,2], Schalk Kok[3], Pedro M. Scheel Monteiro[1]
[1]SOCCO, Council for Scientific and Industrial Research, Cape Town, 7700, South Africa
[2]MaRe, Marine Research Institute, University of Cape Town, Cape Town, 7700, South Africa
[3]Department of Mechanical & Aeronautical Engineering, University of Pretoria, Pretoria, 0028, South Africa
*Correspondence to*: Luke Gregor (lukegre@gmail.com)
**Abstract.** Over the last decade, advanced statistical inference and machine learning have been
used to fill the gaps in sparse surface ocean $CO_2$ measurements (Rödenbeck et al. 2015). The
estimates from these methods have been used to constrain seasonal, interannual and decadal
variability in sea-air $CO_2$ fluxes and the drivers of these changes (Landschützer et al. 2015, 2016,
Gregor et al. 2018). However, it is also becoming clear that these methods are converging towards
a common bias and RMSE boundary: *the wall*, which suggests that $pCO_2$ estimates are now
limited by both data gaps and scale-sensitive observations.   Here, we analyse this problem by
introducing a new gap-filling method, an ensemble of six machine learning models (CSIR-ML6
version 2019a), where each model is constructed with a two-step clustering-regression approach.
The ensemble is then statistically compared to well-established methods. The ensemble,
CSIR-ML6, has an RMSE of 17.16 µatm and bias of 0.89 µatm when compared to a test-dataset
kept separate from training procedures. However, when validating our estimates with independent
datasets, we find that our method improves only incrementally on other gap-filling methods. We
investigate the differences between the methods to understand the extent of the limitations of
gap-filling estimates of $pCO_2$. We show that disagreement between methods in the South Atlantic,
southeastern Pacific and parts of the Southern Ocean are too large to interpret the interannual
variability with confidence. We conclude that improvements in surface ocean $pCO_2$ estimates will
likely be incremental with the optimisation of gap-filling methods by (1) the inclusion of
additional clustering and regression variables (*e.g.* eddy kinetic energy), (2) increasing the
sampling resolution. Larger improvements will only be realised with an increase in $CO_2$
observational coverage, particularly in today's poorly sampled areas.



## 1 Introduction

The ocean plays a crucial role in mitigating against climate change by taking up about a third of the
anthropogenic carbon dioxide ($CO_2$) emissions (Sabine et al. 2004; Khatiwala et al., 2013; McKinley et al.
2016). While the mean state in the global contemporary marine $CO_2$ uptake is a widely-used benchmark (Le
Quéré et al., 2018), underlying assumptions and limited confidence regarding the variability and long-term
evolution of this sink persist. Sparse observations of surface ocean $CO_2$ during winter and in large inaccessible
regions has been the biggest barrier in constraining the seasonal and interannual variability of global
contemporary sea-air exchange (Monteiro et al. 2010; Rödenbeck et al. 2015; Bakker et al. 2016; Ritter et al.
2017). The increasing ship-based sampling effort and the ongoing development of autonomous observational
platforms (e.g. biogeochemical Argo floats and Wave Gliders) have improved confidence of interannual
estimates of ocean $CO_2$ uptake in more recent years (Monteiro et al. 2015; Bakker et al. 2016; Gray et al., 2018).
The community has turned to models and data-based approaches to improve estimates of $CO_2$ uptake by the
oceans for periods and regions with poor or no observational coverage (Wanninkhof et al. 2013a; Rödenbeck et
al. 2015; Verdy and Mazloff, 2017). Ocean biogeochemical models are able to capture the general global trend
in increasing oceanic $CO_2$ uptake shown by observations but suffer from significant regional and interannual (~1
PgC yr$^{-1}$) differences in their estimates because these models cannot yet accurately parameterise the marine
carbonate system at computationally feasible resolutions (Wanninkhof et al. 2013a). In recent years, data-based
approaches, namely statistical interpolations and regression methods, have become a popular alternative to
biogeochemical models (Lefèvre et al. 2005; Telszewski et al. 2009; Landschützer et al. 2014; Rödenbeck et al.
2014; Jones et al. 2015; Iida et al. 2015). The regression methods try to maximise the existing ship-based
observations extrapolating $CO_2$ using proxy variables (observable from space or interpolated). Extrapolating
with proxy variables is possible due to the non-linear relationship between the partial pressure of $CO_2$ ($pCO_2$) in
the surface ocean and proxies that may drive changes in surface ocean $pCO_2$. Improved access to quality
controlled ship-based measurements of surface ocean $CO_2$ through the Surface Ocean $CO_2$ Atlas (SOCAT)
database, and satellite and reanalysis products as proxy variables has aided the development of the data-based
methods (Rödenbeck et al. 2015; Bakker et al. 2016).

### The current state of machine learning in ocean $CO_2$ estimates

With the increase in the number of statistical estimates of surface-ocean $CO_2$, the Surface Ocean $CO_2$ Mapping
(SOCOM) community consolidated fourteen of these methods in an intercomparison of "gap-filling" methods
(Rödenbeck et al. 2015). The intercomparison gives an overview of the SOCOM landscape, with regression and
statistical interpolation approaches making up eight and four of the fourteen methods respectively (Rödenbeck et
al. 2015). Two model-based approaches were also compared.
While SOCOM intercomparison did not identify an optimal mapping method, it weighted the ensemble
members according to how well they represented interannual variability (IAV) relative to climatological surface



ocean $p$CO$_2$ increasing at the rate of atmospheric CO$_2$ concentrations (R$^{iav}$). Two methods, the Jena-MLS
(Mixed-Layer Scheme) and MPI-SOMFFN (Self-Organising Map Feed-Forward Neural-Network) were
weighted more due to lower R$^{iav}$ scores. The MPI-SOMFFN (Self-Organising Map Feed-Forward
Neural-Network), is a global implementation of a two-step clustering-regression approach and has subsequently
become the most widely used method in the literature ( Landschützer et al. 2015, 2016, 2018, Ritter et al. 2017).
The elegance of the clustering-regression approach, particularly the clustering step, is that it reduces the problem
into smaller parts with more coherent variability and reduces the computational size of the problem per cluster –
a beneficial attribute when using regression methods that do not scale well to big datasets.
The SOCOM intercomparison found that the gap-filling methods were in agreement in regions with a large
number of seasonally-resolving persistent measurements, but the different methods did not agree in regions
where data were sparse (e.g. the Southern Ocean).

**1.2 Measuring the uncertainty of estimates?**

The biggest limitation in assessing gap-filling methods is the paucity of data in the Southern Hemisphere
(Rödenbeck et al. 2015; Bakker et al. 2016). The standard use of RMSE and bias as measures of uncertainty
weight the regions or periods with observations heavily compared to the data-sparse regions and periods. The
R$^{iav}$ score improves on the standard implementation of RMSE and bias by weighting the uncertainties annually,
thus giving a less temporally biased estimate of uncertainty. However, the method is still limited to the regions
where there are observations of $p$CO$_2$.
Previous studies have compared their methods' estimates to independent datasets, where measurements of $p$CO$_2$
are not included in the SOCAT datasets (Landschützer et al. 2013, 2014; Jones et al. 2015; Denvil-Sommer et al.
2018). These data serve as good validation data, particularly with the inclusion of derivations of $p$CO$_2$ from
autonomous platforms in the Southern Ocean, a historically undersampled area especially during winter (Boutin
and Merlivat 2013; Gray et al. 2018).
One of the concluding statements in the SOCOM intercomparison is that pseudo- or synthetic data
(deterministic model output) experiments should be used to test and compare methods. Gregor et al. (2017) did
just this, but their study was limited to the Southern Ocean, and the synthetic data did not fully capture the
variability represented by observations, in part due to coarse synthetic data resolution (5-daily mean and ½°
spatially). Moreover, such studies can only compare the limitations of the gap-filling methods within the
framework of the model. The authors found that the ensemble average of the compared methods outperformed
individual methods, in agreement with ensemble approaches previously used in ocean CO$_2$ studies (Khatiwala et
al. 2013).



### 1.3 Aims

The main aim of this study is to present and evaluate a new machine learning approach to estimate surface ocean $pCO_2$. We propose the use of an ensemble, where we hypothesise that the "whole is greater than the sum of its parts" as the strengths of the ensemble members are often complementary in such a way to overcome the weaknesses (Khatiwala et al. 2013; Gregor et al. 2017). Further, we aim to evaluate the method for a selection of existing gap-filling methods. From this comparison we aim not only to gain a sense of our method's performance but also the state of gap-filling based estimates; i.e. where would we be able to improve in future work?

## 2 Methods

There are two major components to this study: surface $pCO_2$ mapping with multiple methods, robust error estimation from SOCAT v5 gridded product and independent data sources. This study takes a similar two-step approach used in the JMA-MLR and MPI-SOMFFN approaches, where data is grouped or clustered first, and then a regression algorithm is applied to each group or cluster. We use the ocean $CO_2$ biomes by Fay and McKinley (2014) as an option for grouping. Alongside this grouping, we use an optimal K-means clustering configuration. Next, four non-linear regression methods are applied to each of the groupings. The regression methods are Support Vector Regression (SVR), Feed-Forward Neural Network (FFN), Extremely Randomised Trees (ERT) and Gradient Boosting Machine (GBM). The latter two approaches are new to the application. These methods are then compared to independent data sources. This is outlined in more detail in the Experimental Overview below.

### 2.1 Experimental overview

The experimental design, outlined below, is summarised in Figure 1:

1. In the first step (described by the "K-means clustering" section in Figure 1), we generate climatological clusters using the oceanic $CO_2$ biomes by Fay and McKinley (2014), and a selection of features variables (five combinations) and number of clusters (a range of clusters from 11 to 25, stepping by two) resulting in a total of 41 clustering configurations.

2. Four regression algorithms are applied to each clustering configuration, resulting in 164 models (described by the "Regression" section in Figure 1). The test data (isolated from model training procedure) is used to identify the best performing cluster with annually weighted bias, root-mean-squared error (RMSE) and $R^{iav}$. The four regression models for $CO_2$ biomes and the four models from the best performing cluster and (as indicated by the bold lines in Figure 1) are used in the steps that follow. The selected eight models are averaged to create an ensemble that is included with the eight members for further evaluation.

3. The third step (as represented by the "K-fold testing" section in Figure 1 and Section 2.5) provides a robust uncertainty evaluation based on the training data (SOCAT v5). An iterative test-train approach



128   is applied to estimate the bias, RMSE and $R^{iav}$ for the complete SOCAT v5 dataset (rather than just one

129   test split).

130  4. The fourth step compares the ensemble estimates of surface ocean $p$CO$_2$ with independent test data

131   (that is not in SOCATv5, as represented by the "Independent" section in Figure 1), which allows testing

132   the predictive skill of the ensemble method (Section 2.6). Four methods from the SOCOM gap-filling

133   intercomparison study are included for reference.

134  5. Lastly, all gap-filling methods are compared to identify regions where there is a divergence in the trend

135   and seasonal cycle.





**Figure 1:** A flow diagram that shows the experimental procedure used in this study. Abbreviations for feature-variables in the orange hexagons can be found in Table 1. All other abbreviations are given in the diagram. Details of each step are given in the text.






**2.2 Data: clustering, training and predictive**

Standard machine learning implementation requires a training- and predictive dataset. The training dataset
consists of a target variable that is being predicted (in this case $pCO_2$) and one or more feature-variables that
have samples that correspond with target samples (*e.g.* SST, Chl-*a*, MLD co-located in space and time), where
feature-variables may directly or indirectly influence the target variable. Features variables are used to predict
once a machine learning model has been trained and must thus be available for the full prediction domain.

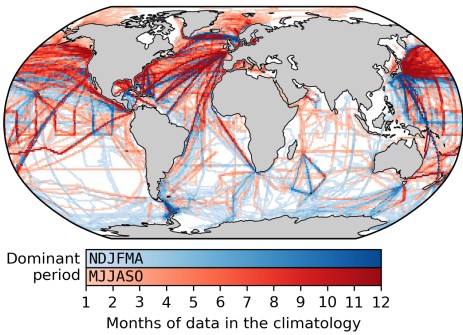

**Figure 2:** Map showing the distribution of the SOCAT v5 monthly gridded product (1982 to 2016) as a monthly climatology
to show how well the seasonal cycle is represented (regardless of the year). The red shading shows grid-points where the
majority of data occur from May to October and the blue shading shows grid-points where the majority of data occur from
November to April.
Here we use surface ocean $pCO_2$ calculated from the SOCAT v5 monthly gridded $fCO_2$ (fugacity of $CO_2$)
product (hereinafter SOCAT v5 as shown in Figure 2) as the target variable (Sabine et al. 2013; Bakker et al.
2016). SOCAT v5 is a quality controlled dataset that contains observations of surface ocean $fCO_2$, which is
converted to $pCO_2$ with:

$$pCO_2 = fCO_2 \cdot exp(\, P_{atm}^{surf} \cdot \tfrac{B + 2 \cdot \delta}{R \cdot T}\, )^{-1}$$

Eq. 1

where $P_{atm}^{surf}$ is the atmospheric pressure at the surface of the ocean, $T$ is the sea surface temperature (SST) in
°K, $B$ and $\delta$ are virial coefficients, and $R$ is the gas constant (Dickson et al. 2007). We used SST from the
Operational Sea Surface Temperature and Sea Ice Analysis (OSTIA) product by GHRSST (Dolon et al. 2012)
and ERA-interim $P_{atm}^{surf}$ (Dee et al., 2011).
Feature-variables in both the training and predictive datasets are globally gridded products, including satellite
observations, *in-situ* measurements and reanalysis products (Table 1, see Section S1 for details). All
feature-variables are gridded to a monthly frequency onto a global 1° × 1° resolution grid. Thereafter, data
processing steps are applied as shown in Table 1 and described in detail in Supplementary Materials (Section
S1) with the final output being a complete dataset ranging from 1982 to 2016. Note that the clustering and
regression steps use different subsets of the feature-variables as indicated in Table 1.





**Table 1**: Summary of the products, variables and data processing steps used for feature-variables. The column "Usage"
indicates the features that are used for the clustering step (identified by C) and for the regression step (identified by R).
Abbreviations are used in Figure 1 and throughout the text. Basic data processing is described in the text with details in the
supplementary materials (Section S1).

| Group: Product | Variable | Abbrev | Usage | | Processing | Reference |
|---|---|---|---|---|---|---|
| GHRSST: **OSTIA** | Sea surface temperature | SST | C | R | - | Donlon et al. (2012) |
| | SST seasonal anom. | SST' | C | R | *SST – annual average* | |
| | Sea ice fraction | ICE | | R | - | |
| MetOffice: **EN4** | Salinity | SSS | | R | - | Good et al. (2013) |
| CDIAC: **ObsPack v3** | Atmospheric $pCO_2$ | $pCO_2^{atm}$ | | R | *$xCO_2^{atm}$ × sea level pressure* | Masarie et al. (2014) |
| UCSD: **Argo Mixed Layers** | Mixed Layer Depth | MLD | C | R | $\log_{10}(climatology)$ | Holte et al. (2017) |
| ESA: **Globcolour** | Chlorophyll-*a* | Chl-*a* | C | R | $\log_{10}(climatology\ filled_{1982-1997}^{cloud\ gaps})$ | Maritorena et al. (2010) |
| | Chla seasonal anom. | Chl-*a*' | | R | *Chl-a – annual average* | |
| ECMWF: **ERA-Interim 2** | *u*-wind | *u* | | R | - | Dee et al. (2011) |
| | *v*-wind | *v* | | R | - | |
| | Wind speed | $U_{10}$ | | R | $\sqrt{u^2 + v^2}$ | |
| ESA: **Globcurrent** | Eddy kinetic energy | EKE$^{clim}$ | C | | $\log_{10}(½ \cdot (u'^2 + v'^2))$ | Rio et al. (2014) |
| - | Day of the year | *J* | | R | $\sin(\frac{j}{365})$, $\cos(\frac{j}{365})$ | - |
| LDEO: **$pCO_2$ climatology** | Surface ocean $pCO_2$ | $pCO_2^{clim}$ | C | | Data smoothing | Takahashi et al. (2009) |

In this paragraph, we briefly describe the data processing steps shown in Table 1 - detailed product descriptions
and in-depth processing steps are in Section S1. We derive an additional SST feature, SST′, by subtracting the
annual mean of SST from each respective year, leaving the annual mean anomalies (Donlon et al. 2012). We use
the $\log_{10}$ transformation of the Globcolour Chl-a global product (Maritorena et al. 2010). Cloud gaps and the
period before the start of the product (1982 to 1997) are filled with the climatology (1998 – 2016), and
high-latitude winter regions (where there is no climatology for Chl-*a*) is filled with low concentration random
noise. We derive an additional Chl-a feature, Chl-*a*′ using the same procedure as described for the SST annual
mean anomalies. We use a $\log_{10}$ transformation of mixed layer depth (MLD) from Argo float density profiles
(Holte et al. 2017) to create a monthly climatology, thus imposing the assumption that there is no interannual
variability. Wind speed is calculated from 6-hourly data using the equation in Table 1 before taking the monthly
average. Atmospheric $pCO_2$ is calculated with: $pCO_2 = xCO_2^{atm} \times P^{atm}$, where $xCO_2^{atm}$ is the mole fraction of
atmospheric $CO_2$ (from ObsPack v3 by Masarie et al. 2014) and $P^{atm}$ is reanalysed mean sea-level pressure
(from ERA-interim 2; Dee et al. 2011) – further details for the procedure are in the Section S1 of the
Supplementary Materials. The climatology of Eddy Kinetic Energy (EKE$^{clim}$) is calculated from *u* and *v* surface
current components (integrated for depth < 15 m) from the Globcurrent product (Rio et al., 2014), where $u'$ is
calculated as $\bar{u} - u$ and similarly with *v* (Table 1).




### 2.3 Clustering and biomes

The seasonal and interannual variability of global surface ocean $p$CO$_2$ is complex due to interactions of various
driver variables acting on the surface ocean at different space and time scales (Lenton et al. 2012; Landschützer
et al. 2015; Gregor et al. 2018). Machine learning algorithms applied globally struggle to represent the $p$CO$_2$
accurately unless spatial coordinates are included as feature-variables (Gregor et al. 2017). A common practice
is to divide the ocean into regions where processes that drive $p$CO$_2$ are coherent and then apply regressions to
each region – five of the eight regression methods in Rödenbeck et al. (2015) apply this approach.  We adopt
two approaches to develop regions of internal coherence in respect of CO$_2$ variability.
Our first "clustering" approach uses the oceanic CO$_2$ biomes by Fay and McKinley (2014) that divide the ocean
into 17 biomes. Fay and McKinley (2014) define their biomes by establishing thresholds for SST, Chl-*a*, sea-ice
extent and maximum MLD depth. Unclassified regions from the original biomes are manually assigned based on
their geographical extent resulting in six additional regions (Figure 3). Note that we may refer to the modified
Fay and McKinley (2014) ocean CO$_2$ biomes as CO$_2$ biomes from here on. For later analyses, we group certain
biomes together as shown by the brackets above the colour-bar in Figure (3).

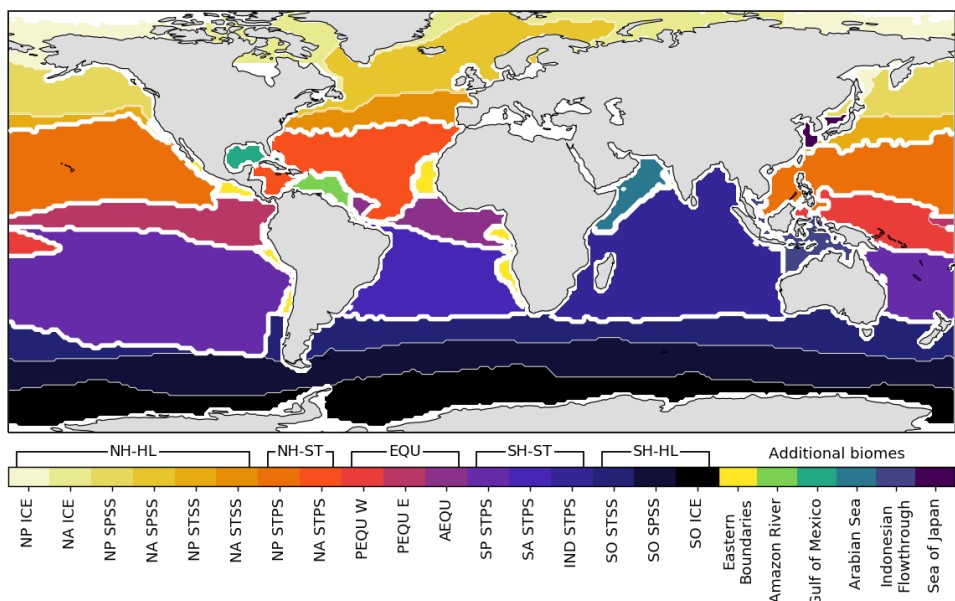

**Figure 3:** Regions or biomes as defined by Fay and McKinley (2014). Unclassified regions from the original data have been
assigned manually in this study and are shown by the separate colour palate. This modified configuration of the CO$_2$ biomes
is referred to as BIO23 in this study. The sea-mask used in Lanschützer et al. (2014) has been applied. For the biome
abbreviations (below the colour-bar) see Fay and McKinley (2014). The abbreviations above the colour-bar are used in this
study, where selected biomes are grouped together. Thick white lines show the boundaries of the grouped regions. Prefixes
are: NH = Northern Hemisphere, SH=Southern hemisphere; suffixes are HL = high latitudes, ST = subtropics, and EQU =
equatorial.





Further, we also use K-means clustering, specifically the mini-batch K-means implementation in Python's
Scikit-Learn package (Sculley 2010; Pedregosa et al. 2012), which is described in the supplementary materials
(Section S2.2; Figure S2). We apply clustering with various feature combinations and the number of clusters
(shown by orange hexagons in Figure 1). We tested the number of clusters ranging from 11 to 25 (stepping by
two). The performance of each cluster is not tested with a clustering metric; instead, we test the performance
based on the test scores of the regressions in the next step as a more complete indicator of performance. We find
optimal results in respect of RMSE and biases with 21 and 23 clusters (Figure 5). We selected 21 clusters
(Figure S2). Each method of defining regional coherence in respect of $p$CO2 variability has its methodological
weaknesses so in this study we adopted the approach of incorporating both K-means and $CO_2$ biomes into the
ensemble (Figure 1).  Although this likely weakens the geophysical meaning of the ensembled domains we
show that it strengthens the overall performance of the ensemble (Figure 5).

**2.4 Regression**

Here we describe the underlying machine learning principles of regression (*a.k.a.* supervised learning). The
co-located data (*i.e.* SOCAT v5) are split into training and test-subsets with a roughly 80:20 split. The test-subset
is isolated from the training process to attain a reliable estimate of uncertainty. We make the split between
training and test-subsets based on a random subset of years in the time series (1982 to 2016): 1984, 1990, 1995,
2000, 2005, 2010 and 2014. We avoid using a shuffled train–test split (completely random) as this leads to
artificially low uncertainties in machine learning algorithms that are prone to overfitting (see the experiment in
S2.1), where the models can reproduce the shuffled test data better as these data are adjacent to samples of the
same ship track.
Machine learning models have the ability to be as complex as the dataset at hand and are thus at risk of fitting
not only the signal but also the noise of the training data – this is known as the bias-variance tradeoff. High
variance is a result of a machine learning model that is too complex and is fitting the noise, and high bias is due
to insufficient complexity where the model cannot fit the signal (Hastie et al. 2009). Machine learning
algorithms have hyper-parameters that control the complexity of the model for each specific problem. In this
study, hyper-parameters are tuned by training the model with grid-search cross-validation, where a portion of the
training subset is iteratively kept separate from the training process for a certain set of hyper-parameters. The
hyper-parameters that result in the best score from the grid-search are used for the fit with the full training
subset. We use a variation of K-fold cross-validation called *group K-fold* in Scikit-Learn (Pedregosa et al. 2012).
Rather than having arbitrary splits for each fold, a given grouping variable is used to split the data – in this case,
years. Using years as the grouping variable reduces bias towards the second half of the time series where data is
less sparse.
The train-test split and cross-validation are applied identically to each of the four machine learning algorithms
for each clustering configuration. We use the following machine learning algorithms: Extremely Randomised
Trees (ERT – Geurts 2006); Gradient Boosting Machines (GBM – Friedman 2001); Support Vector Regression



(SVR – Drucker et al. 1997); and Feed-Forward Neural Networks (FFN). The details of these methods and how
they were tuned are explained in the supplementary materials (Section S2.3). The first two methods, ERT and
GBM, are new to this application. SVR has been implemented as a single global domain by Zeng et al. (2017),
and FFN is used by several different methods, some of which are in the SOCOM intercomparison (Landschützer
et al. 2014; Zeng et al. 2014; Sasse et al. 2013).
Regression performance is tested using RMSE primarily but also bias (Equations 3 and 4) and $R^{iav}$ (Equation 5)
with only the models from the best averaged cluster used for the rest of the study.

### 2.5 Robust biases and root-mean-square errors

Standard practice in machine learning is to set aside a test-subset of the data as described in Section 2.4. We use
this standard approach in the second step of our experiment as an estimate of the performance for each of the
machine learning models (164 in total). However, this grouped train-test split gives a bias and RMSE estimate
limited to the random test years of test-subset (see Section 2.4). To overcome this limitation, we apply the
train-test split method five times in a K-fold-like test approach (Figure 1: "K-fold testing" section), meaning that
the data in a test fold is never used to train the model. The splits in the test fold are also based on a subset of
years spaced five years apart. We then refactor the five test-fold estimates into a complete test-estimate (with the
same structure as the original SOCAT v5), thus giving a complete estimate of bias and RMSE. This robust
test-estimate method ensures that correct biases and RMSE scores are reported even if methods are prone to
overfitting (see Section S2.1 and Figure S1). We limit this procedure to only the $CO_2$ biome and best cluster
regressions as it has five times the computational cost of a single train-test split.

### 2.6 Method validation data

For method validation we use observation data that are not used in SOCAT (Figure 4 and Table 2) as they are
either: 1) included in LDEO, but not SOCAT; 2) not measured with an infrared analyser; 3) derived from two
other variables in the marine carbonate system, where these include dissolved inorganic carbon (DIC), pH and
total alkalinity (TA) – SOCCOM floats use empirically calculated TA.

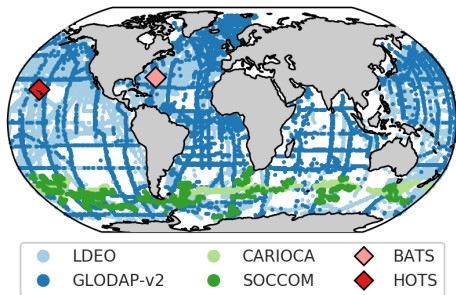

**Figure 4:** The distribution of the validation data. Details of these datasets are given in Table 2. HOTS and BATS are marked
as diamonds to distinguish them as time series stations.



**Table 2:** Details for the validation datasets. The measured variables are shown (DIC = dissolved inorganic carbon; TA = total
alkalinity) along with the estimated accuracy of $p$CO$_2$. This includes the propagated uncertainty in the conversion from DIC
and TA to $p$CO$_2$ as defined by Lueker et al. (2000), where the estimates marked with * are an extrapolation of the estimates
as the DIC and TA uncertainties do not match or exceed those listed in the publication. Grid points show the number of data
at the same resolution as the feature-variables.

| Platform | Project | Measured variable | Accuracy (µatm) | Reference | Grid points |
|---|---|---|---|---|---|
| Ship | LDEO | $p$CO$_2$ Equilibrator | ±2.5 µatm | Takahashi et al. (2016) | 16161 |
| | GLODAP v2 | DIC + TA | ~ 12 µatm @ 400 µatm * | Olsen et al. (2016) | 5976 |
| Surface floats | CARIOCA | $p$CO$_2$ Colourimetric | ±3.0 µatm | Boutin and Merlivat (2013) | 613 |
| Profiling floats | SOCCOM | pH + TA (LIAR) | ~ 11 µatm @ 400 µatm | Carter et al. (2016) | 1037 |
| Mooring | BATS | DIC + TA | ~ 4 µatm @ 400 µatm | Bates (2007) | 246 |
| | HOTS | DIC + TA | < 7.6 µatm @ 400 µatm * | Dore et al. (2009) | 214 |

The uncertainty of $p$CO$_2$ that is calculated from DIC and TA is dependent on the accuracy of these two
measurements, as well as the derivation of $p$CO$_2$ with dissociation constants, for which we use the *CBSys*
package in Python (Hain et al. 2015). *CBSys* implements the constants from Lueker et al. (2000) that reports an
uncertainty of 1.9% standard deviation of the calculated $p$CO$_2$ where DIC and TA uncertainties are 2.0 and 4.0
µmol.kg$^{-1}$ respectively. The measurements in GLODAP v2 are slightly larger than this at 4 and 6 µmol.kg$^{-1}$,
which would result in an error larger than 1.9% – this is 12 µatm for a 400 µatm estimate at a hypothetical 3%
error. While this potentially large error range may seem concerning, we argue that the inclusion of these data in
data-sparse regions is more valuable than their omission. Moreover, the errors from the previous gap-filling
products are on the order of 20 µatm, below the potential uncertainty from the DIC/TA conversion to $p$CO$_2$
(Landschützer et al. 2014; Rödenbeck et al. 2014). Williams et al. (2017) estimated the error for $p$CO$_2$ calculated
empirically to be 2.7%, where TA was calculated empirically with the Locally Interpolated Alkalinity
Regression (LIAR) algorithm (Carter et al. 2016). All $p$CO$_2$ data are then gridded to the same time and space
resolution as the feature-variables (monthly × 1°) using *xarray* and *pandas* packages in Python (McKinney,
2010; Hoyer and Hamman, 2017).

**2.7 Sea-air CO$_2$ flux calculation**

Sea-air CO$_2$ flux ($F$CO$_2$) is calculated with:

$$FCO_2 = K_0 \cdot k_w \cdot (pCO_2^{sea} - pCO_2^{atm})$$

Eq. 2

where $K_0$ is the solubility of CO$_2$ in seawater (Weiss 1974) and $k_w$ is the gas-transfer velocity calculated from
wind speed using formulation by Wanninkhof et al. (2013). We scale $k_w$ so that the global mean is 16 cm hr$^{-1}$,
following the same procedure as Landschützer et al (2014). $p$CO$_2^{sea}$ is from the gap-filling methods, and $p$CO$_2^{atm}$
is atmospheric $p$CO$_2$. All ancillary variables required in these calculations are the same as those listed in Table 1,
except for $p$CO$_2^{atm}$, which is the CarboScope atmospheric $p$CO$_2$ product from Rödenbeck et al. 2014.

**2.8 Relative interannual variability and interquartile range metrics**

**2.8.1 Regression metrics**
We use bias and root-mean-square error (RMSE) as first-order metrics of model performance.



Bias is the mean difference between the target variable and the estimates thereof:

$$Bias = \sum_{i=1}^{n} \frac{\hat{y}_i - y}{n}$$

Eq. 3

where $n$ is the number of training samples, $y$ is the array of target data and $\hat{y}$ is the corresponding array of
estimates. Similarly, RMSE is a measure of the difference between the target variable and the estimates thereof:

$$RMSE = \sqrt{\sum_{i=1}^{n} \frac{(y_i - \hat{y}_i)^2}{n}}$$

Eq. 4

In our study, these metrics are calculated for each year and then the mean of the annual bias or RMSE scores is
taken as a more robust measure of performance in the context of temporally imbalanced data. This is typically
done for the global domain unless otherwise stated.
The relative interannual variability metric ($R^{iav}$) was introduced by Rödenbeck et al. (2014) and used in the
SOCOM intercomparison by Rödenbeck et al. (2015) to measure how well a method represents the interannual
variability of SOCAT v5. The metric furthers the idea of RMSE calculated by year (and region if stated,
otherwise global) by normalising annually weighted RMSE to a benchmark with minimal interannual and
seasonal variability:

$$R^{iav} = \frac{\sigma_{1982-2015}(M^{iav\,(t)})}{\sigma_{1982-2015}(M^{iav\,(t)}_{bench})}$$

Eq. 5.1

$$M^{iav\,(t)} = \sqrt{\frac{\sum_{i=1}^{n}(y_i - \hat{y}_i)}{n-1}}$$

Eq. 5.2

$$M^{iav\,(t)}_{bench} = \sqrt{\frac{\sum_{i=0}^{n}(y_i - \hat{y}_i^b)}{n-1}}$$

Eq. 5.3

Here $\sigma$ is the standard deviation of $M^{iav}$ and $M^{iav}_{bench}$ respectively, which are both represented as yearly time
series. Equations 5.2 and 5.3 show the formulation for $M^{iav\,(t)}$ and $M^{iav\,(t)}_{bench}$, which represent these metrics for a
single year. The symbol $i$ represents individual data points in a particular year $t$, $y$ is the observation-based data
for that year, $\hat{y}$ is the predicted data and $n$ is the number of points in the year and region. The benchmarked
$M^{iav}_{bench}$ is calculated to normalise $M^{iav}$. The $\hat{y}^b$ represents the data that has been corrected for IAV by subtracting
the climatology and atmospheric $p\text{CO}_2$ trend from the predictions.

### 2.8.2 Ensemble metrics

We use the interquartile range (IQR) between different gap-filling methods as a robust metric of disagreement,
where the standard deviation is sensitive to outliers. IQR is calculated as the third quartile (75th percentile) minus
the first quartile (25th percentile). The disagreement between methods is calculated with interannually resampled
data and then averaged over the time series to arrive at the interannual disagreement (IQR$^{IA}$). This is calculated
per pixel if the representation of the data is spatial (maps) and per time step of a time series.






## 3 Results

### 3.1 Regression results
The results from the second part of the experiment (as shown in Figure 1) are depicted in Figure (5a-c) which
plots the matrix of the (a) average bias, (b) RMSE and (c) $R^{iav}$ for each combination of the experimental number
of clusters and clustering features. The RMSE and bias are calculated by averaging the annual estimates for the
randomly selected test years (as explained in Section 2.4) rather than using the entire dataset - this is done to
minimise the effect of the temporal imbalance in the number of observations.

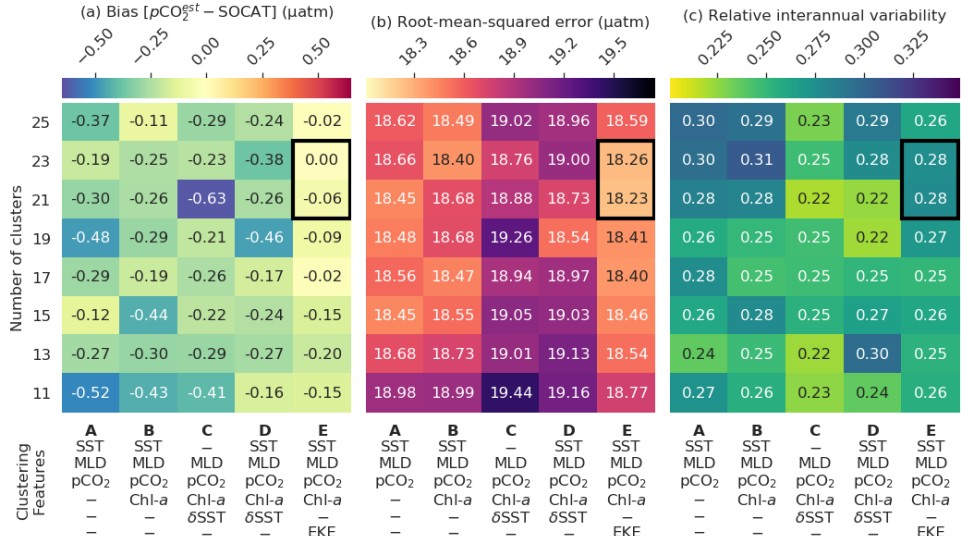

**Figure 5:** Heatmaps showing the average cluster (a) bias, (b) root-mean-squared error (RMSE) and (c) relative interannual
variability ($R^{iav}$) for different cluster configurations, where smaller scores are better for all metrics. The rows show the
number of clusters, and the columns show clustering feature-variable configurations. Each cluster contains the average of
scores for four regression methods: support vector regression, extremely randomised trees, gradient boosting machine, and
feed-forward neural-network. The black box indicates clustering configurations that perform well across all metrics – note
that a $R^{iav} < 0.3$ falls within the best category of performance in Rödenbeck et al. (2015).
Results show that the configuration that includes $EKE^{clim}$ (column E in Figure 5a-c) as a clustering feature has
the lowest average RMSE and absolute bias for nearly all clusters, regardless of the number of clusters (rows in
Figure 5a,b). The increased dynamics associated with high EKE regions might change the way $pCO_2$ behaves
compared to low EKE regions (Monteiro et al. 2015; du Plessis, 2017, 2019). The optimal number of clusters
within this configuration is either 21 or 23, based on the smallest bias and RMSE scores (as indicated by the
black box in Figure 5). Note that we do no weight $R^{iav}$ strongly in this assessment as a $R^{iav}$ score of less than 0.3
is in the top performing category in the SOCOM intercomparison (Rodenbeck et al. 2015). We select the
configuration with the lowest RMSE, which has 21 clusters with the following features: SST, $\log_{10}(MLD^{clim})$,





$p$CO$_2$$^{clim}$, log$_{10}$(Chl-$a^{clim}$), and log$_{10}$(EKE$^{clim}$); and is hereinafter abbreviated as K21E (see Figure S2 for the
distribution of the climatology for these clusters).
Comparatively, the Fay and McKinley (2014) CO$_2$ biomes have an average RMSE score of 18.98 μatm (Table 3)
but have a lower mean R$^{iav}$ (0.26) and smaller bias (0.03 μatm) than the K21E configuration. Given that the CO$_2$
biomes perform well and provide an alternate clustering approach, we include the regression estimates
(hereinafter we refer to the Fay and McKinley (2014) CO$_2$ biomes with the six additional biomes as BIO23).
The eight machine learning models from K21E and BIO23 (four each) were used to create an ensemble by
averaging $p$CO$_2$ estimates (CSIR-ML8).
**Table 3:** Regression scores for the CO$_2$ biomes (BIO23), the cluster configuration from column E in Figure 5 (K21E) and the
ensemble (CSIR-ML8). Abbreviations are: RMSE = root-mean-square error; R$^{iav}$ = relative interannual variability (Equation
5). Regression methods are: SVR = support vector regression; ERT = extremely randomised trees; GBM = gradient boosting
machine; FFN = feed-forward neural-network. Bold values are significantly lower than the mean for that column ($p < 0.05$
for two-tailed $Z$-test; absolute values used for bias column).

| Cluster | Regression | Bias (μatm) | RMSE (μatm) | R$^{iav}$ |
|---------|-----------|-------------|-------------|-----------|
| **CSIR-ML8** | | **0.04** | **17.25** | 0.25 |
| **K21E** | SVR | -0.45 | **17.95** | 0.24 |
| | ERT | 0.84 | **17.96** | 0.36 |
| | GBM | -0.32 | 18.21 | 0.24 |
| | FFN | -0.30 | 18.82 | 0.27 |
| **BIO23** | SVR | **-0.19** | 18.47 | **0.15** |
| | ERT | 0.85 | 18.76 | 0.38 |
| | GBM | **0.02** | 19.05 | 0.28 |
| | FFN | -0.58 | 19.65 | **0.21** |

All regression methods have lower RMSE scores for K21E than for BIO23, but R$^{iav}$ and bias do not indicate that
any of the two clustering approaches is preferable (Table 3). Comparing the RMSE scores of the individual
regression methods, we see that the model scores are ranked the same in each cluster from first to last: SVR,
ERT, GBM, FFN. However, it is important to note that this ranking does not apply to bias or R$^{iav}$, where ERT has
low RMSE, but the largest bias and R$^{iav}$ in each cluster. CSIR-ML8 outperforms nearly all its members with
RMSE and bias scores of  17.25 μatm and 0.04 μatm respectively. However, the ensemble R$^{iav}$ (0.25) is only just
less than the average of the ensemble members' average (0.26).

**3.2 Robust RMSE, bias and R$^{iav}$**

Here, we study the change in the bias and RMSE for all selected methods (i.e. K21E, BIO23 and CSIR-ML8;
Table 3) across 1982-2016 (Figure 6). Most notable is that bias scores for all models have the same interannual
tendencies, with a positive bias at the beginning of the time series (1982 to 1993) that is strongest before 1990,
strongly influencing the mean bias (Table 4). Secondly, the biases for K21E (solid lines) are, on average, smaller





than for BIO23 (dashed lines) as shown for the annually averaged results in Table 4 (0.73 µatm and 2.24 µatm
respectively). These biases are much larger than those reported in Table 3 (with averages of absolute biases of
0.48 µatm and 0.41 µatm for K21E and BIO23 respectively), but this is likely since selected test years (black
triangles in Figure 6b) fall on years of low bias. While FFN has the largest RMSE (18.93 µatm and 20.24 µatm
for K21E and BIO23), it has a smaller bias compared to other regression methods (0.04 µatm and 1.60 µatm
respectively), motivating for including FFN regressions in the ensemble (Table 4). Conversely, the ERT
approach has a significant positive bias (2.08 µatm and 3.88 µatm for K21E and BIO23 respectively, with $p >$
0.95 for both values; Table 4). A second ensemble without ERT regressions, thus with six members
(CSIR-MLR6 version 2019a, hereafter called CSIR-ML6), has lower biases compared to CSIR-ML8 (0.98 µatm
and 1.48 µatm respectively; Table 4).

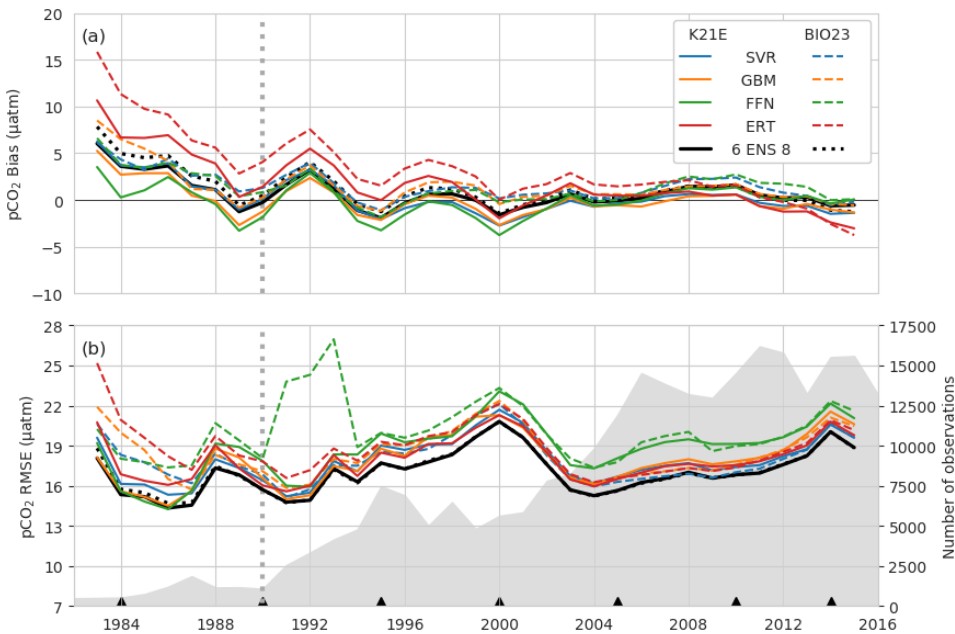

**Figure 6:** Annually averaged (a) bias and (b) RMSE for the eight individual regression methods in Table 3: BIO23 (dashed
lines) and K21E (solid lines). The dotted black lines show the ensemble averages for all eight models (CSIR-ML8), and the
solid black line shows metrics for the ensemble of the SVR, GBM and FFN (CSIR-ML6) from BIO23 and K21E. The grey
filled area in (b) shows the number of observations per year and black triangles shows the years that are isolated as the test
subset. The vertical dashed grey line demarks 1990 prior to which there is a large positive bias.
Similarly to the biases, RMSE for all models (Figure 6b) have similar interannual tendencies and variability,
with a sharp peak in the year 2000 ( > 20 µatm where the mean RMSE is 18.61 µatm). The increased RMSE
scores are likely due to the spatial distribution of sampling (see Figure S3), *e.g.* an increase in sampling in the
high latitudes during spring and summer, a region and period of high variability and biogeochemical complexity,
would increase the weight of these data in the final RMSE calculation, thus resulting in larger RMSE scores.
The increase in the number of samples from 2002 to 2016 results in a sharp decrease in RMSE ( < 19 µatm for
the majority of this period). Both ensembles outperform all other methods for the majority of the time series





with RMSE scores of 17.16 µatm and 17.25 µatm for CSIR-ML6 and CSIR-ML8 respectively (see Table S1
comparisons of ensembles with different members).
The $R^{iav}$ scores for the robust errors (Table 4) are lower than train-test results with a single split reported in Table
3, likely due to an increase of standard deviation for the IAV benchmark (Equation 5). The lowest score is held
by CSIR-ML6 (0.20) and is lower (better) than the average for its members (0.21). These $R^{iav}$ estimates compare
well to the Jena-MLS and SOM-FFN, which both scored < 0.3 (Rödenbeck et al. 2015).
**Table 4:** The robust estimates of bias, RMSE and $R^{iav}$ from 1982 to 2016 for BIO23, K21E and the ensemble averages,
CSIR-ML6 and CSIR-ML8, where the first excludes the ERT. Bold values are significantly lower than the mean for that
column ($p < 0.05$ for two-tailed Z-test; absolute values used for bias column). See Table S1 for further comparisons between
different ensemble configurations.

| Cluster | Regression | Bias (µatm) | RMSE (µatm) | $R^{iav}$ |
|---|---|---|---|---|
| **CSIR** | **ML6** | 0.98 | **17.16** | **0.20** |
| | **ML8** | 1.48 | **17.25** | 0.22 |
| **K21E** | **SVR** | **0.58** | 18.04 | 0.21 |
| | **ERT** | 2.08 | 18.20 | 0.27 |
| | **GBM** | **0.21** | 18.05 | **0.21** |
| | **FFN** | **0.04** | 18.93 | 0.22 |
| **BIO23** | **SVR** | 1.76 | 18.17 | 0.21 |
| | **ERT** | 3.88 | 19.16 | 0.32 |
| | **GBM** | 1.72 | 18.59 | 0.21 |
| | **FFN** | 1.60 | 20.24 | **0.21** |

The spatial distribution of the bias and RMSE is now studied for the CSIR-ML6 ensemble (Figure 7 a and b,
respectively), particularly focusing on the regional patterns emerging from the data. CSIR-ML6 clearly
represents the subtropical regions (NH-ST and SH-ST) with relatively low biases and RMSE scores (< |5 µatm|
and 10 µatm respectively). The equatorial regions (EQU), especially the eastern Pacific, contrasts this with large
uncertainties in both bias and RMSE (> |10 µatm| and 30 µatm respectively). The high-latitude oceans (NH-HL
and SH-HL) have considerable uncertainties due to the large interannual variability of surface ocean $p$CO$_2$
caused by the formation and retreat of sea-ice (around Antarctica; Ishii et al. 1998; Bakker et al. 2008) and
phytoplankton spring blooms (Atlantic sector of the Southern Ocean, North Pacific and Arctic Atlantic;
Thomalla et al. 2011; Lenton et al. 2013; Gregor et al. 2018). There are two bands of overestimates on the
southern and northern boundaries of the North Atlantic Gyre, where the latter coincides with the Gulf Stream.
Regression approaches may be prone to a positive bias in the North Atlantic as this was also shown by
Landschützer et al. (2013; 2014).





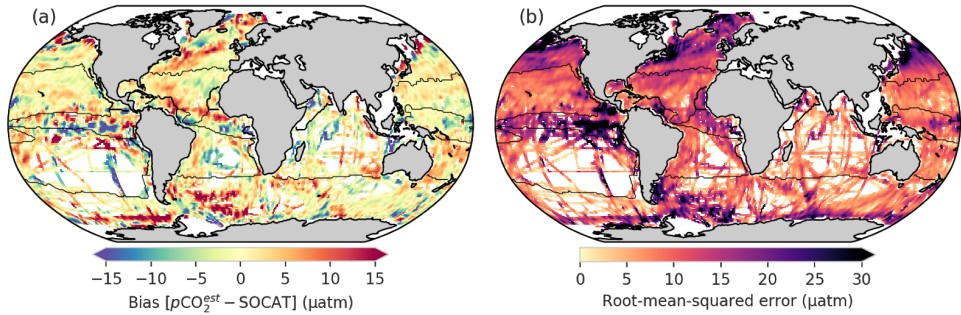

**Figure 7:** (a) shows the biases from the robust test-estimates; (b) shows the root-mean-squared errors for CSIR-ML6. A convolution has been applied to (a) and (b) to make it easier to see the regional nature of the biases and RMSE. Figure S4 shows the bias for every ensemble member.

In summary, the robust test-estimates show that there is a bias positive bias in $pCO_2$ predictions before 1990 for all models, but is largest for ERT and excluding these models from the ensemble results in better $pCO_2$ predictions. The spatial evaluation of the performance metrics for CSIR-ML6 shows that regions with specific oceanic features (e.g. western boundary currents) mostly have positive biases. However, it is important to note that these uncertainty assessments are limited as the characteristics and biases of the dataset are intrinsic to the models. Validation with independent data is thus a more reliable estimate of the performance of these methods.

### 3.3 Validation with independent datasets

Here, we validate the accuracy of $pCO_2$ estimates from CSIR-ML6 with independent data (that is not in SOCAT v5 as described in Table 2). To further study the behaviour of our ensemble estimates relative to previous studies, we compare the results from four independent methods of the SOCOM intercomparison project against the independent data (Rödenbeck et al. 2015). Those four independent methods are: the Jena mixed-layer scheme (Jena-MLS version *oc_v1.6*, Rödenbeck et al. 2014); Japanese Meteorological Agency – multi-linear regression (JMA-MLR updated on 2018-12-2, Iida et al. 2015); Max Planck Institute – Self-organising Map Feed-forward Neural-network (MPI-SOMFFN *v2016*, Landschützer et al. 2017); and University of East Anglia – Statistical Interpolation (UEA-SI version 1.0, Jones et al. 2015). $pCO_2$ estimates by the Jena-MLS were resampled to monthly temporal resolution and interpolated to a one-degree grid using Python's *xarray* package.

The performance of each gap-filling method is represented with a Taylor diagram for each independent validation dataset (Figure 8; Taylor et al. 2001). The most important characteristic learnt from these plots is that the gap-filling methods are tightly bunched for nearly all validation datasets, indicating a similar RMSE, correlation and standard deviation relative to the reference datasets. Poor estimates in Figures 8a-d may indicate that the training data for gap-filling methods is the limiting factor. Secondly, the gap-filling methods almost always underestimate the standard deviation of the validation datasets, being below the black arced line for all but HOTS (Figure 8e).



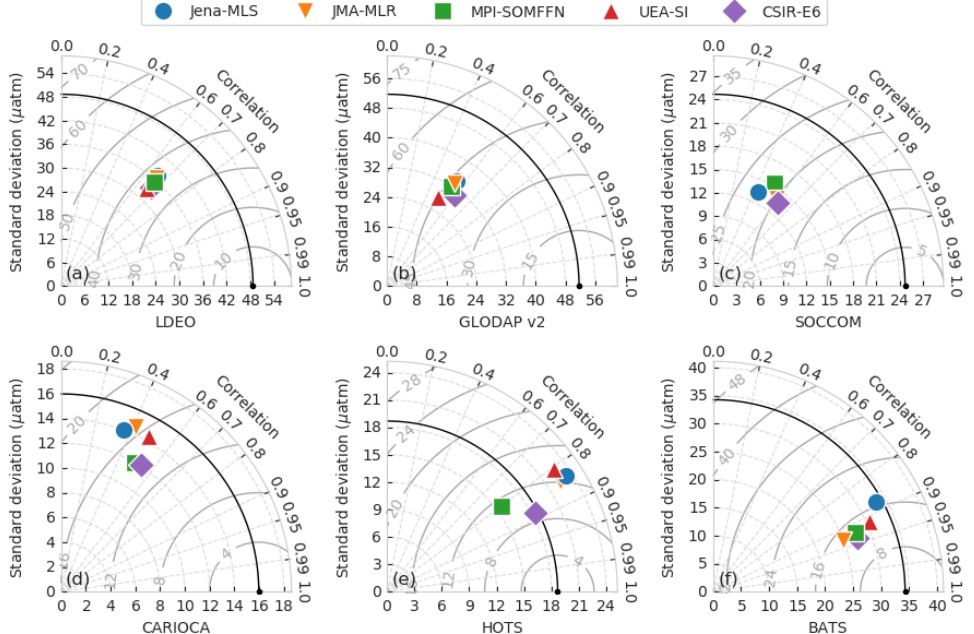

**Figure 8:** Taylor diagrams comparing the $pCO_2$ estimates of five gap-filling methods with validation datasets (Table 2), for the period 1990-2015. Each validation dataset has its own Taylor diagram as labelled on the bottom axes. The black marker on the bottom axis in each subplot represents the validation dataset and the black arc shows the standard deviation thereof. The closer that the gap-filling estimates are to this point, the better the model's performance, in terms of variance, centred RMSE and correlation (for bias information, see Table 5). The solid grey arcs show the centred RMSE for the datasets (with bias removed).

All methods fail to represent the standard deviation of the two global validation datasets, LDEO and GLODAP v2 (Figures 8a,b), with centred RMSE scores greater than 35 µatm. However, calculating RMSE annually results in scores of ~27 µatm for LDEO and ~35µatm for GLODAP v2, much lower than shown in Figure 8a,b due to high RMSE scores (> 40 µatm) for a small subset of years (Section S3.3 and Figure S54). Estimates of the Southern Ocean datasets (Figures 8c, d), SOCCOM and CARIOCA, have lower RMSE scores (~16 µatm and ~23 µatm respectively) relative to LDEO and GLODAP v2. However, for standard deviation scores of similar magnitude and low correlation coefficients, the datasets are not well constrained (Table 5). The SOCCOM dataset also has the largest average absolute bias for estimates, with gap-filling methods underestimating by at least 11 µatm (Table 5). This large bias may be because SOCCOM floats have a proportionately large number of winter samples – suggesting that our knowledge of Southern Ocean winter fluxes are largely underestimated (Williams et al. 2017). In contrast, all methods estimate the two time-series stations, HOTS and BATS (Figures 8e,f and Table 5) relatively well with correlation scores of > 0.8 and low average bias ~4.5 µatm.



**Table 5:** The RMSE and bias for each gap-filling method compared to the validation datasets. For more information on the
validation-datasets see Table 2. The first row of data (count) shows the number of gridded samples in the dataset during the
period 1990-2015 (that are not in the SOCAT v5 gridded product). Values shown in bold are significantly different from the
mean for the column ($p < 0.05$ for two-tailed $Z$-test; absolute values used for biases).

| Metric | Method | LDEO | GLODAP-v2 | SOCCOM | CARIOCA | BATS | HOTS |
|--------|--------|------|-----------|--------|---------|------|------|
| **Count** | **Count** | 16161 | 5976 | 1037 | 613 | 246 | 214 |
| **RMSE** | **CSIR-ML6** | **26.55** | **32.84** | 23.15 | **14.26** | **12.53** | **8.62** |
| | **MPI-SOMFFN** | 27.43 | 35.96 | 25.21 | 15.08 | 13.39 | 10.40 |
| | **JMA-MLR** | 29.11 | 34.53 | **22.32** | 16.05 | 14.29 | 11.64 |
| | **Jena-MLS** | 27.61 | 35.52 | 26.83 | 18.24 | 16.14 | 12.28 |
| | **UEA-SI** | 27.35 | 35.07 | | 15.73 | 13.35 | 18.52 |
| **Bias** | **CSIR-ML6** | -1.18 | 8.48 | -13.12 | 4.28 | **0.32** | 0.46 |
| | **MPI-SOMFFN** | **-0.19** | 9.16 | -13.79 | 4.00 | -1.41 | -0.12 |
| | **JMA-MLR** | -1.86 | **6.62** | **-11.25** | 2.85 | -3.98 | 2.22 |
| | **Jena-MLS** | **-0.14** | 8.48 | -14.68 | 7.18 | 4.09 | 6.15 |
| | **UEA-SI** | -0.71 | 9.20 | | **0.79** | -2.02 | 16.27 |

Despite all scores being closely grouped (Figure 8), Table 5 shows that the CSIR-ML6 method scores
significantly lower RMSE scores (using a two-tailed $Z$-test with $p < 0.05$) for all but one of the datasets
(SOCCOM). However, bunching of the RMSE scores (Figure 8) is beneficial with regard to achieving low
$p$-values. No single method dominates the biases, with JMA-MLR and MPI-SOMFFN each scoring the lowest
bias on two occasions. To summarise, all gap-filling methods underperform when validated against independent
observational products. Tight bunching of gap-filling method scores per validation dataset shows that training
data may limit all methods in the same manner.

**3.4 The effect of uncertainties on the sea-air CO$_2$ flux interannual variability**

In this section, we assess the regional implications of the differences in gap-filling methods' estimates of the
sea-air CO$_2$ flux ($F$CO$_2$) over the period 1990 to 2016. $F$CO$_2$ was calculated using the same gas transfer velocity
and solubility for each gap-filling method (Section 2.7). Differences in $F$CO$_2$ are thus driven by variations in
$p$CO$_2$ from each gap-filling method.



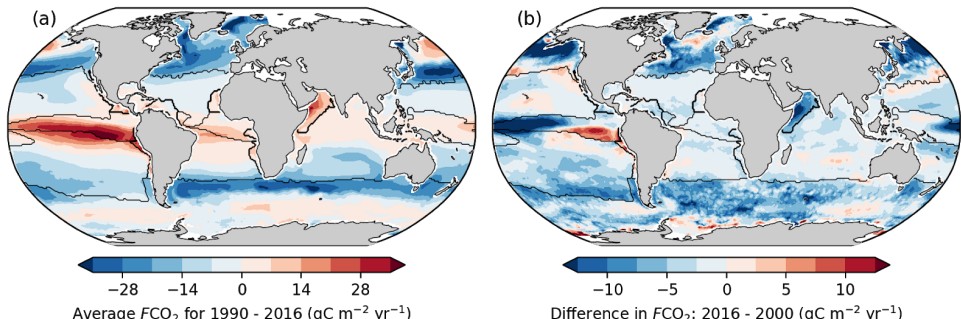

**Figure 9:** (a) Average sea-air $CO_2$ fluxes ($FCO_2$) of CSIR-ML6 for 1990 to 2016, where $FCO_2$ is calculated as shown in Equation 2. Negative $FCO_2$ (blue) indicates regions of atmospheric $CO_2$ uptake. (b) The difference between $FCO_2$ in 2016 and 2002, which are the minimum and maximum of global ocean uptake flux ($FCO_2$) estimates respectively (for CSIR-ML6 in Figure 10a). Black lines show the regions as defined in Figure 2.

The average $FCO_2$ for 1990-2016 by CSIR-ML6 (Figure 9a) contextualises the regional distribution of fluxes: strong outgassing in the Equatorial Pacific, strong sink in the mid-latitudes, a moderate uptake for the most part of the subtropics, and weak source in the majority of the Southern Ocean (in agreement with e.g. Takahashi et al., 2009). The global annual time-series for $FCO_2$ as simulated by CSIR-ML6 (Figure 10a) indicates a strengthening for 2000 to 2016 (as for the other methods). To give spatial context to this strengthening, we display the differences in $FCO_2$ between 2016 and 2000 (Figure 9b), since those are the two years where the difference in global $FCO_2$ is greatest for CSIR-ML6 (Figure 10a). Note that Figure 9b serves as a snapshot for the change in $FCO_2$ between those two years, whose interpretation cannot be linked to an overall anthropogenically-forced change as the comparison between two years could highlight interannual, decadal or multi-decadal variability. The differences in $FCO_2$ between 2016 and 2000 is negative in the high latitudes and moderately positive in the subtropics, indicating a respective increase and decrease in the $CO_2$ ocean uptake between the two years. The Eastern Equatorial Pacific is the only region that shows a considerable increase in $FCO_2$ (> 10 gC m$^{-2}$ yr$^{-1}$) between the two specific years.

The annual change in $FCO_2$ is also studied for the different regions. The Southern Hemisphere high-latitude (SH-HL) region is the strongest contributor to the trend (Figure S6b), where there is a steady increase in the uptake of $CO_2$ since the 2000s for all methods (Landschützer et al. 2015; Gregor et al. 2018). On average, the Northern Hemisphere high latitudes (NH-HL) are a weaker sink relative to the SH-HL, because the SH-HL is more than double the area of the NH-HL (Figure S6c). The equatorial (EQU) region is the only persistent source of $CO_2$ to the atmosphere (also seen in Figure 9a). The subtropical regions (Figure 10c, e) contribute to global flux on similar orders of magnitude; however, there is a large divergence between gap-filling methods in the SH-HL.





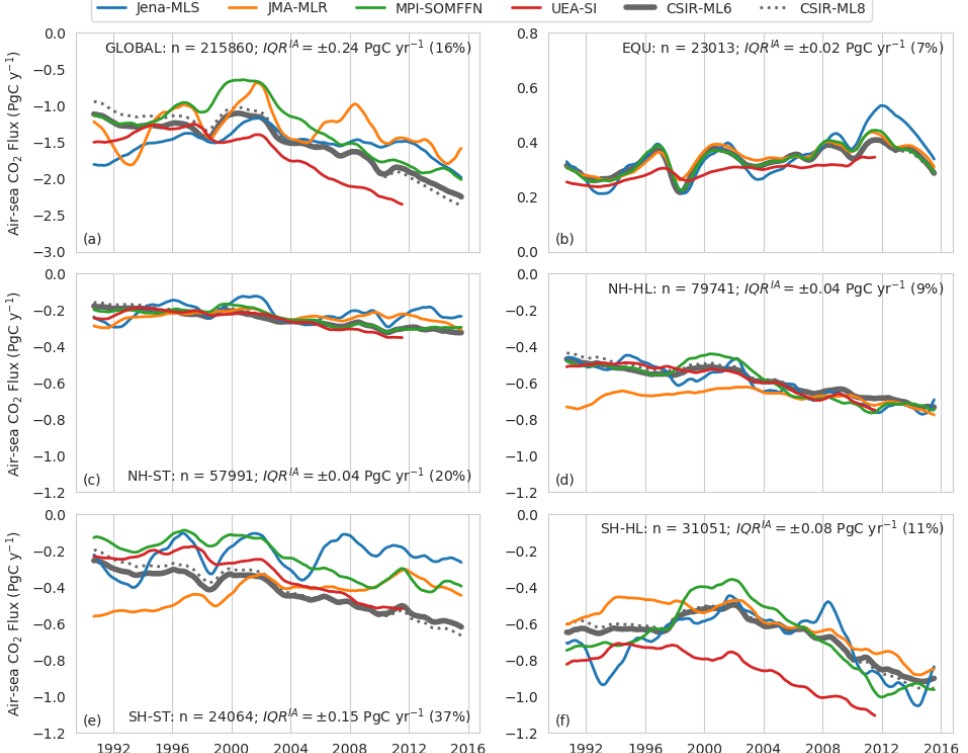

**Figure 10:** Sea-air $CO_2$ fluxes averaged for regions as shown in Figure 2: (a) global domain, (b) Equatorial regions, (c) Northern Hemisphere Subtropical, (d) Northern Hemisphere High Latitude, (e) Southern Hemisphere Subtropical. (f) Southern Hemisphere High Latitude. The coloured lines show the four SOCOM products. The thick and dotted grey lines show the results for CSIR-ML6 and CSIR-ML8, respectively. A moving average of 12 months has been applied to smooth the data. Note that the y-axes scales differ for the top (a) and (b). The text at the right of each figure shows the number of SOCAT v5 gridded data points for each region ($n$) and the inter-annual interquartile range ($IQR^{IA}$).

We use the average interquartile range between the one-year rolling mean estimates ($IQR^{IA}$) as a measure of agreement or divergence between gap-filling methods, where large values indicate a divergence (Section 2.8.2). We also show the $IQR^{IA}$ scaled to the range of the regional interannual variability (max – min) as a percentage (relative $IQR^{IA}$), which shows if the trend for a particular region is agreed on by all methods (the smaller the percentage, the better the agreement across methods). The disagreement between methods in the SH-ST is substantial (Figure 10e), with diverging $F CO_2$ throughout the period with an $IQR^{IA}$ of 0.15 PgC yr⁻¹ and a very large relative $IQR^{IA}$ of 37%. Similarly, the $IQR^{IA}$ for the SH-HL region (Figure 10f) is 0.08 PgC yr⁻¹, but the relative $IQR^{IA}$ is much lower at 11%, indicating that all methods agree on the observed strong trend. Compared to the Southern Hemisphere, the Northern Hemisphere regions are both relatively well constrained, with $IQR^{IA}$ estimates of 0.04 PgC yr⁻¹ for both regions (Figure 10c,d). However, a large relative $IQR^{IA}$ of 20% suggests that the interannual $F CO_2$ estimates in this region are potentially not resolving the trend, or more likely that there is a





weak trend with a small difference between the minimum and maximum interannual estimates of $F\mathrm{CO_2}$. The
equatorial region (EQU - Figure 10b) has the lowest $\mathrm{IQR^{IA}}$ and relative score at 0.02 PgC yr$^{-1}$ and 7%.
The CSIR-ML8 method is not included in the $\mathrm{IQR^{IA}}$ calculations but is included in Figure 10 to show the impact
of the ERT models' positive bias in $p\mathrm{CO_2}$ on $F\mathrm{CO_2}$ (Figure 6a). The biases are positive at the beginning and
negative end of the time series, with the average absolute difference between the CSIR methods being 0.08 PgC
yr$^{-1}$. The positive biases have the strongest impact in the SH-ST that occupies 36% total area (Figure S6c), with
only 11% of the total observations in SOCAT, suggesting that this method is sensitive to imbalanced datasets.

**3.5 Regional disagreement between methods**

In order to better understand the regional distribution of the uncertainties in $F\mathrm{CO_2}$, we assess the level of
agreement between methods in their interannual surface ocean $p\mathrm{CO_2}$ estimates (Figure 11). We use $p\mathrm{CO_2}$ for this
representation as no spatial integration occurs – only time averaging.

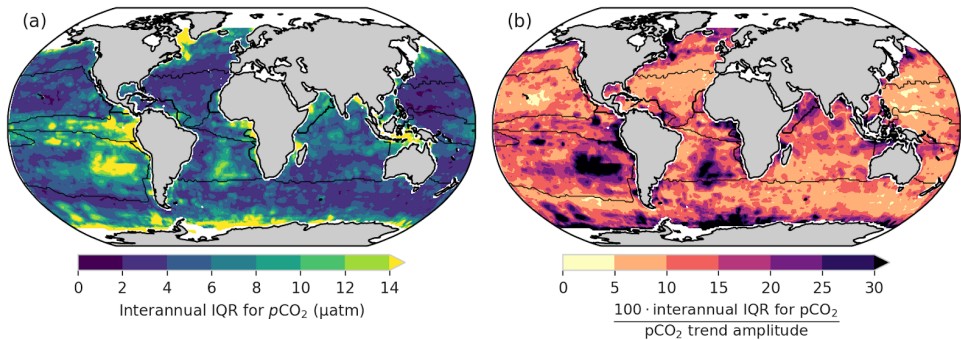

**Figure 11:** (a) The magnitude of the interannual disagreement between gap-filling methods ($\mathrm{IQR^{IA}}$). (b) Level of agreement
on the interannual variability across methods, more specifically $\mathrm{IQR^{IA}}$ scaled by the difference between the maximum and
minimum values for interannual $p\mathrm{CO_2}$ (the range).
The interannual estimates of interquartile range ($\mathrm{IQR^{IA}}$; Figure 11a) show the disagreement between methods is
relatively small in the majority of the ocean ($\approx 5$ µatm); the exceptions being the South Atlantic, southeastern
Pacific and eastern equatorial Pacific with differences of $> 10$ µatm. The $\mathrm{IQR^{IA}}$ scaled to the
maximum-minimum range of interannual $p\mathrm{CO_2}$ suggests that the NH-ST is well constrained ($< 10\%$), which is
in conflict with the $\mathrm{IQR^{IA}}$ for $F\mathrm{CO_2}$ in Figure 10c (where the relative $\mathrm{IQR^{IA}}$ is 20%). The disagreement may stem
from the magnifying impact that wind speed has on $F\mathrm{CO_2}$, *i.e.* small differences in $p\mathrm{CO_2}$ may become large
when fluxes are calculated. The same principle may apply to the EQU in Figure 11b, where relative $\mathrm{IQR^{IA}}$ is
large ($> 10\ \%$) for $p\mathrm{CO_2}$, but low wind speeds result in a low relative $\mathrm{IQR^{IA}}$ for $F\mathrm{CO_2}$ (7% in Figure 10b). The
largest relative $\mathrm{IQR^{IA}}$ scores occur in the SH-ST ($ > 10\%$ in Figure 11c) where data is sparse, specifically the
South Atlantic and southeastern Pacific (Figure 2a). The relative $\mathrm{IQR^{IA}}$ scores suggest that the gap-filling
methods agree on $p\mathrm{CO_2}$ in the SH-HL east of the Greenwich meridian ($> 0°$ E).



In summary, we show that there is an agreement between gap-filling methods in the Northern Hemisphere for
interannual $p$CO$_2$, but the methods show considerable disagreement in the Southern Hemisphere, particularly in
the subtropics. Disagreements in the Equatorial and Southern Hemisphere high-latitude regions are large (>
10%) and should be treated with caution when considering trends in these regions.

## 4 Discussion

### 4.1 Not all models are equal

In their study, Khatiwala et al. (2013) stated that: "*our comparison of different methods suggests, that multiple*
*approaches, each with its own strengths and weaknesses, remain necessary to quantify the ocean sink of*
*anthropogenic CO$_2$*". In our study, we embrace this philosophy by creating an ensemble of two-step machine
learning models that estimate global surface ocean $p$CO$_2$. The authors of the SOCOM intercomparison
(Rödenbeck et al. 2015) warn against the use of ensembles with the statement: *"We also discourage any*
*ensemble averaging (or medians, etc.) of full spatiotemporal fields or time series, as this would result in*
*variations that are not self-consistent any more and fit the data less well than individual products".* Our
approach may seem in opposition to the statement, but we show robustly that the CSIR-ML6 method reproduces
the available data with greater accuracy than previous methods, albeit in an incremental way. Our method is
methodologically consistent with regard to feature-variables. Though there is variability in the clustering and the
regression, we create the ensemble with a good understanding of each model's biases (Figure 6 and Figure S4).
The argument that ensembles reduce transparency is also somewhat diminished by the fact that little additional
information that can be gained from highly non-linear models, with the exception of basic diagnostics such as
feature-variable importance (see Figure S7) from decision-tree-based approaches (Pedregosa et al. 2012;
Castelvecchi, 2016). Our results thus show that there is, in fact, a benefit in creating an ensemble of models
(Table 5), and if carefully implemented is an additional tool that can be used to reduce the uncertainties in
gap-filling estimates of $p$CO$_2$.
It could be argued that an exhaustive search for the optimal configuration (Figure 5) for CSIR-ML6 may result
in poorly trained individual models. However, we think that the merit of introducing and assessing regression
algorithms new to the application (for gradient boosting machines and extremely randomised trees) outweighs
the marginal loss in potential performance for individual methods. Moreover, lessons learnt from our study can
be used to improve on future iterations. It also makes the case for ensembles stronger as the CSIR-ML6
performs well relative to other gap-filling methods.
In the search for the optimal clustering configuration (Figure 5a,b), we show that including EKE (along with
SST) as a clustering feature-variable leads to an improvement in bias and RMSE for nearly all number of
clusters. Increased intra-seasonal variability of $p$CO$_2$ appears to be associated with regions of high EKE
compared to low EKE regions (Monteiro et al. 2015; du Plessis, 2017, 2019). Moreover, the importance of EKE





as a part of the cluster constraints also shows that more thought should be given to how we sample $pCO_2$ in
high-EKE regions and at what resolution regression methods are run at – we discuss this in detail later.
Our findings suggest the following about the individual regression methods: the SVR and GBM algorithms
produce good estimates with lower RMSE scores and biases, the FFN approach has larger RMSE scores yet low
biases than the other methods, and the ERT approach has low RMSE scores but large biases in the estimates
(Figure 6a,b; Table 4). We do not include the ERT approach in the ensemble (CSIR-ML6) due to the large
time-evolving biases, suggesting that ERT (with our tuning) is not suitable for estimating surface ocean $pCO_2$.
The bias in ERT may be due to its sensitivity to imbalanced datasets (Crone and Finlay, 2012), where the data in
SOCAT v5 are few before 2000. Returning to the above quote by Khatiwala et al. (2013), we thus find that the
weaknesses of ERT outweigh its strengths.

### 4.2 Divergent gap-filling estimates

While we see that the improvements in the performance of gap-filling methods are relatively stagnant (relative
to the training and validation data), the differences between the methods' estimates of $pCO_2$ and $FCO_2$ vary
significantly in some regions particularly in regions where data is sparse such as in the Southern Hemisphere
oceans (Figure 2). We also find that training the gap-filling methods with limited training data exposes the
intrinsic biases of the algorithms, or in the words of Ritter et al. (2017): "*the difference [between ga-filling*
*methods] is a result of how the spatial and seasonal heterogeneity and the sparseness of the data is dealt with*".
Conversely, as the number of training data increase, the biases are reduced, and the methods converge.

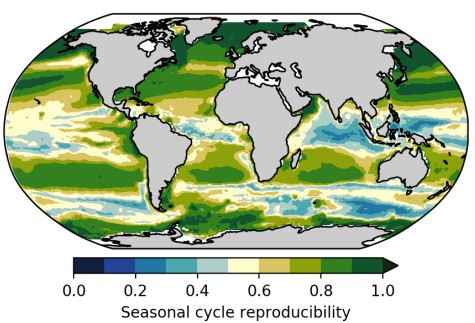

**Figure 12:** The seasonal cycle reproducibility of CSIR-ML6 $pCO_2$, which is a correlation of detrended $pCO_2$ with its own
climatology – the larger the correlation the stronger the reproducibility of the seasonal cycle (method from Thomalla et al.
2011).

The Northern Hemisphere subtropical regions are a good example of a region where the gap-filling methods
converge (Figure 11b), as also shown by the low RMSE scores and high correlation for the two mooring
stations, HOTS and BATS (Figure 8e,f). One of the reasons that the methods can predict the variability well in
the subtropics (Figure 8e,f) is because these regions are less biogeochemically complex and driven primarily by
seasonal changes in SST (Bates 2001; Dore et al. 2009). This strong SST-driven seasonality in the subtropics is
shown by the high seasonal cycle reproducibility (Figure 12).



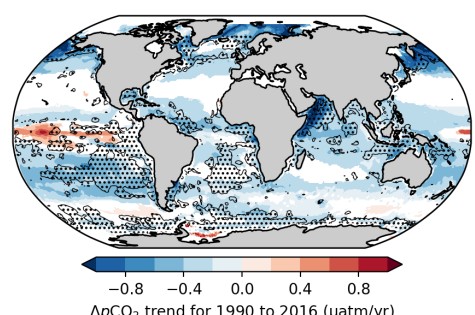

$\Delta pCO_2$ trend for 1990 to 2016 (µatm/yr)

**Figure 13:** $\Delta pCO_2$ trends ($p < 0.05$), where $\Delta pCO_2$ is calculated as the estimated surface ocean $pCO_2$ from the
CSIR-ML6 method minus atmospheric $pCO_2$ from the CarboScope project (Rödenbeck et al. 2014). The shaded areas
show the regions where $IQR^{IA}$ is > 15%, thus indicating regions where trends should be interpreted with caution.

The gap-filling methods' divergences also serve as a metric to inform where there is not enough data to

constrain the $pCO_2$ or $FCO_2$ estimates, *i.e.* the divergences inform us where estimates should be treated with

caution. The $IQR^{IA}$, when scaled to the range of the interannual variability (Figure 11b), should be taken into

account when analysing interannual trends of $\Delta pCO_2$ (Figure 13). For instance, trend estimates in $\Delta pCO_2$  for

CSIR-ML6 are negative ($p < 0.05$) for the majority of the global ocean, even in regions where method estimates

are too disparate to resolve interannual variability (relative $IQR^{IA}$ > 15%; Figure 13). However, the relative

$IQR^{IA}$ is not without its limits, as there may be regions where methods are in agreement but share the same

biases, thus reporting false confidence in the estimates. Regions of false confidence would most likely occur in

data sparse areas, but could only truly be identified with better data coverage in these regions.

### 4.3 Inching up and over the wall: incremental improvements

In our study, we show that all gap-filling methods suffer from the same uncertainties where there are data to test

and validate the estimates (Figure 8), and divergences between estimates when there are insufficient data to

constrain the methods (Figure 11b). From these points, it may seem that we may have in fact "hit the wall" in

terms of better resolving surface ocean $pCO_2$. In this section, we discuss how we might overcome this proverbial

wall. First, by first addressing the uncertainty and biases within the methods, and then discussing the issue of

data scarcity, specifically, how could we most effectively improve our sampling strategies to close the gaps in

the current datasets.

### 4.3.1 Reducing systematic errors

The robust test-estimates show that there are regions where training data is not sparse, yet estimates still suffer

from large uncertainties (*e.g.* northern and southern boundaries of the North Atlantic gyre in Figure 7a,b and

Figure S4). These errors are spatially consistent with those reported by Landschützer et al. (2014). Such regional

mismatches between gridded observations and estimates are likely systematic – meaning that gap-filling

methods are not able to resolve the more complex $pCO_2$ variability at current resolutions (monthly × 1° or



coarser) or with the current regression feature-variables (Gregor et al. 2017; Denvil-Sommer et al. 2018). It may
be possible to reduce these uncertainties with consideration about the drivers of $CO_2$ in a specific region.
Including appropriate additional feature-variables (if available), such as reanalysis mixed-layer depth products,
may improve the uncertainties of gap-filling methods (Gregor et al. 2017). Similarly, increasing the temporal
and spatial resolution may be able to improve estimates where aliasing occurs in regions of high dynamic
variability such as the mid-latitude oceans (Monteiro et al. 2015). It is worthwhile noting that increasing the
resolution may not be the panacea for poor estimates. For example, the Jena-MLS method is able to estimate
$pCO_2$ with relative accuracy (Figure 8) at a low spatial ($\approx 4° \times 5°$; Rödenbeck et al. 2014); however, with the
trade-off in spatial resolution, the method is able to increase the temporal resolution to 6-hourly estimates.
One of the weaknesses of our study is that our approach is similar to other clustering-regression methods,
namely MPI-SOMFFN and JMA-MLR, which could lead to similar biases between these clustering-regression
methods. Importantly, this highlights the need for new methods that are fundamentally different and may lead to
the development of procedural architectures that might be able to resolve the biases in well-sampled regions
better. For example, a recent study by Denvil-Sommer et al. (2018) developed a method (LSCE-FFNN) that first
estimates the climatological $pCO_2$ and then the anomalies from this climatology – their method reported RMSE
scores on the order of those reported in this study (~18.0 μatm) and very low $R^{iav}$ scores (< 0.2). While new
methods might not lead to drastic reductions in uncertainties, incremental improvements in uncertainties will be
driven by approaches that offer new solutions, whether it be increased resolution, additional feature-variables or
a new approach.

### 4.3.2 Scale-sensitive sampling strategies

All gap-filling methods suffer from similar biases and uncertainties (Figure 8, Table 5) when compared to
independent validation data, yet the same methods show vastly different results in data-sparse regions. These
shared uncertainties and regionally-consistent divergences between methods suggest that insufficient training
data is the limiting factor (Rödenbeck et al.2015; Landschützer et al. 2016; Ritter et al. 2017; Denvil-Sommer et
al. 2018). Our study highlights the need for targeted sampling in these data-sparse regions, with the relative
$IQR^{IA}$ metric (Figures 11b) providing a guideline of where sampling should occur to better resolve interannual
$pCO_2$. Large mismatches in the Southern Hemisphere subtropics and the Southern Ocean suggest that these
remote regions require more data to be constrained.
Autonomous sampling platforms, such as biogeochemical Argo floats, surface drifters and wave gliders, are
offering a new and efficient way to target inaccessible regions with relative affordability at the scales required to
resolve not only interannual but also intraseasonal variability (e.g. Monteiro et al. 2015). Despite being
potentially less accurate than the SOCAT requirements, including these measurements might still result in
improved $pCO_2$ estimates as long as measurements are not positively or negatively biased (Wanninkhof et al.
2013b).



While autonomous platforms offer a low-cost solution to improve data coverage in data-sparse regions, there
needs to be a better understanding of the required sampling rates to resolve $pCO_2$ at any given location and
season - scale sensitivity question – a point that also addresses the issue of increasing the resolution of
gap-filling methods. Observing system simulation experiments (OSSEs) offer useful insight into the required
sampling density and frequency (Lenton et al. 2006, Lenton et al. 2009, Majkut et al. 2014; Mazloff et al. 2018;
Kamenkovich et al. 2011, 2017). The majority of these OSSEs have been focussed on resolving fluxes in the
Southern Ocean, which perhaps deserves the attention as it is the largest contributor to interannual $FCO_2$
variability (Figure S6b; Landschützer et al. 2016). Another Southern Ocean study found that a sampling rate of
at least three days was required to resolve intraseasonal variability in a region with high dynamic variability
such as the SH mid-latitude oceans (Monteiro et al. 2015) – a much higher sampling rate than the 10-day period
for carbon (pH)-enabled Argo floats.
Finally, over and above the focus of recent work on the Southern Ocean, there seems to be a gap in the
community's efforts in reducing the uncertainties in the Southern Hemisphere subtropical oceans – a region with
few observations (Figure 2) and significant disagreement between methods (Figure 10). Importantly, the eastern
Pacific and eastern Indian oceans may be more variable than their well sampled Northern Hemisphere
counterparts as suggested by the spatial autocorrelation length-scales of $pCO_2$ (for where there are
measurements) and satellite proxies (SST, Chl-$a$ and sea surface height; Jones et al. 2012). And while the
gap-filling methods estimate that there is high seasonal cycle reproducibility in these regions (Figure 12;
meaning that gap-filling methods might well resolve them), we do not have enough information about the
carbon cycle in these regions to make these assumptions. If anything, this should be an encouragement to the
community that these undersampled regions can easily be resolved, especially with the use of autonomous
sampling platforms.

### 5 Summary

Our study suggests that we may be reaching the limits of gap-filling methods' abilities to reduce uncertainties,
as shown by the limited incremental improvement in errors by the ensemble method we compare with
established methods. Significant uncertainties still prevail across all gap-filling methods, most likely limited by
the extent of basin-scale observational gaps in the Southern Hemisphere as well as sampling aliases in
mesoscale intensive ocean regions. We propose ways in which the surface ocean $CO_2$ community can improve
estimates within the bounds of the current observations, and make recommendations for future observations.
We introduce a new surface ocean $pCO_2$ gap-filling method that is a machine learning ensemble of six two-step
clustering-regression models (CSIR-ML6 version 2019a). An exhaustive search process was used to find the
best K-means clustering configuration which was used alongside the Fay and McKinley (2014) oceanic $CO_2$
biomes. The regression models applied to each clustering method are support vector regression, feed-forward
neural-networks and gradient boosting machines. We show that the ensemble of the six methods outperforms



each of its members, thus promoting the idea that averaging model estimates, each with different strengths and
weaknesses, results in an improvement in the overall estimates.
The CSIR-ML6 (version 2019a) ensemble approach was compared to validation data alongside four other
methods from the SOCOM intercomparison study (Rödenbeck et al. 2015). Our new method marginally
outperformed the SOCOM methods when comparing RMSE scores for the validation data, but fared equally on
biases. Despite this improvement, all methods had errors of roughly the same magnitude, suggesting that the
methods are resolving $pCO_2$ equally outside the bounds of the training data.
Closer assessment of the spatial distribution of errors shows that there is spatial coherence between regression
approaches for the Northern Hemisphere. Some of these errors coincide with regions of high dynamic variability
or complex biogeochemistry, suggesting that increasing the spatial and temporal resolution of gap-filling
methods could improve estimates. Moreover, introducing additional feature-variables for regression, such as
eddy kinetic energy, may improve estimates in these regions.
A comparison of the spatial distribution of mismatches in $pCO_2$ between gap-filling methods shows that there
are regions (primarily in the Southern Hemisphere) where the compared methods, as an ensemble, cannot
resolve interannual variability of $pCO_2$. These large mismatches are likely to occur due to amplification of
methodological biases in data-sparse areas. We propose that scale-sensitive integrated multi-platform sampling
of $pCO_2$ in these regions should be the top priority for the community - a task that is made easier by the
development of autonomous sampling platforms. Moreover, we suggest that optimised simulation sampling
experiments should be used to understand the spatial and temporal requirements of $pCO_2$ in different regions and
periods.
In closing, we suggest that it is time to consider another SOCOM-like intercomparison. Several new methods
have been developed since the last intercomparison and the addition of these would improve the robustness of
ensemble flux estimates. Further, the authors of the SOCOM intercomparison suggest that a future
intercomparison should include a comparison of methods using simulated data, a method to overcome the
limitation of the lack of data to test the estimates.

**Code and data availability**

Supporting code is available in Supplementary Materials. Data (global surface ocean $pCO_2$ from CSIR-ML6
version 2019a) is available at https://doi.org/10.6084/m9.figshare.7894976.v1.

**Author contributions**

LG is the lead author and developed the method and wrote the manuscript. ADL contributed to the model
assessment and contributed in editing the manuscript. SK contributed to the initial conceptualisation of the



methods and proofread the manuscript. PMSM contributed to the development of the manuscript and its
reviews.

**Acknowledgements**

This work is part of a Post-doctoral research fellowship funded by the CSIR's Southern Ocean Carbon - Climate
Observatory (SOCCO) through financial support from the Department of Science and Technology (DST) and
the National Research Foundation (NRF) and hosted at the MaRe Institute at UCT.  We acknowledge the support
from the Centre for High Performance Computing (CSIR-CHPC). The Surface Ocean $CO_2$ Atlas (SOCAT) is an
international effort, endorsed by the International Ocean Carbon Coordination Project (IOCCP), the Surface
Ocean Lower Atmosphere Study (SOLAS) and the Integrated Marine Biogeochemistry and Ecosystem Research
program (IMBER), to deliver a uniformly quality-controlled surface ocean $CO_2$ database. The many researchers
and funding agencies responsible for the collection of data and quality control are thanked for their contributions
to SOCAT.

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
