# Peer review of "A comparative assessment of the uncertainties of global surface-ocean CO2 estimates using a machine learning ensemble (CSIR-ML6 version 2019a) – have we hit the wall?"

_Geoscientific Model Development, 2019_

## Referee Comment (RC1) · Peter Landschützer (Referee) · 24 Apr 2019

Review of Gregor et al: A comparative assessment of the uncertainties of global surface ocean CO2 estimates using a machine learning ensemble (CSIRML6 version 2019a) – have we hit the wall? Submitted to GMDD

Hamburg, April, 24th 2019, Reviewer: Peter Landschützer

Summary:

Gregor and colleagues present an impressive and comprehensive study, comparing the performance of various machine learning-based regression approaches in combination with different ocean-biome combinations. The authors compare their new estimates with the current "state-of-the-art" methods represented in the SOCOM intercomparison project and a wide range of independent and novel validation data. Using this impressive set of data, the authors ask the question, whether we have "hit the wall" in our accuracy to reconstruct the ocean carbon sink, and in what way we may further improve in the future.

Strengths:

I am particularly impressed by the amount of data and experiments used by the authors. And despite this vast amount of information, the manuscript is clearly written and easy to follow. The study provides a step forward compared to other existing intercomparison studies (e.g. the cited papers of Rödenbeck et al 2015 and Ritter et al 2017) as it compares a more consistent set (or ensemble) of estimates, all created within the same regions and with the same observational dataset (SOCATv5). Additionally, I am not aware of any other study that makes use of such a large set of independent estimates (GLODAPv2, SOCCOM, etc.) to validate their results. All of the above are significant steps forward and provide a well-suited set-up for answering the research questions posed by the authors.

Weaknesses:

I have not encountered any major weakness.

Recommendation:

This study is a significant contribution to our scientific understanding of current "state-of-the-art" observation-based pCO2 and air-sea CO2 flux estimates, their limitations in space and time as well as potential pathways for future improvement. The study will instantly be of interest and benefit ongoing carbon cycle assessment studies, such

as the Regional Carbon Cycle Assessment and Processes phase 2 (RECCAP2), the Global Carbon Budget by the Global Carbon Project and last but not least, the IPCC assessment reports. I therefore recommend timely publication of manuscript. In the following, I have only listed a few minor/technical comments to be considered by the authors.

Specific and minor comments to the text:

Page 2 line 61: "did not identify" – I would rather say that SOCOM "did not look for" the best method. Unlike this study, the SOCOM comparison was slightly more difficult as some methods were still based on older observational datasets (e.g. LDEOv1) which made an objective comparison with one dataset (e.g. SOCATv5) additionally challenging.

Page 3 line 72-73: correct. Very little agreement was found in the SO, however, it is worth noting that Ritter et al also found remarkable agreement regarding decadal signals, despite the strong discrepancies in seasonality and IAV.

Page 9 lines 186-187: I find myself arguing a lot against the direct inclusion of spatial coordinates. The reason is that CO2 is not directly affected by longitude or latitude, but rather direct environmental proxies that vary along space and time (e.g. SST). Adding Lat and Lon might decrease the error metric as it replaces an unknown. However, in a process-sense it makes much more sense to apply some special selection by regimes, biomes or clusters.

Page 9 figure 3: I absolutely understand the advantage of adding additional regions to the "blank" Fay and McKinley biomes, but a bit more motivation would be good on how these additional regiosn were chosen (e.g. it is immediately obvious to combine the EBUS regions, but why not the very small Sea of Japan with the surrounding waters?)

Page 14 lines 322-323: This is a repeat and can be removed

Page 14 Figure 5 and following text: The authors in the text often use words like "outperform" (e.g. line 355, line 384) or "much larger" (e.g. line 364). As a reader when I hear outperform, I immediately think of a 50% error reduction or similar, whereas in fact not a single bias value in Figure 5 or table 3 qnd table 4 are above $1\mu$atm. Just to put this into context: The current flag A measurement uncertainty is in the range of $1$-$2\mu$atm. Hence these differences are small. They might be significant, but certainly don't deserve wording like "outperformed"

Page 18 line 410: word "bias" occurs twice – remove first occurance

Page 24 lines 544-546: One has to be careful quoting the SOCOM study here. The SOCOM study faced 2 additional differences which made a combination trickier. Firstly, not all methods were based on the same observational dataset (some were built on 1 million data, others on almost 15 million at the time). Secondly, not all methods did span the same geographical extend (with some covering more of the coast, less of the Arctic, etc.). Based on all these differences (and others), Rödenbeck et al 2015 avoided to provide an ensemble mean estimate.

Page 24 and onward: The authors argue for the inclusion of EKE to move forward, but I was not fully convinced, given the small improvement in Figure 5. Another limiting factor that is not discussed is availability. In an ideal case one would use time varying fields of SST, SSS, DIC, TALK, etc. however, in most cases they are not available. As the authors mention, EKE only exists as a climatology, hence one cannot expect directly improved IAV signals from it (as e.g. visible in Figure 5c). I do however agree with the authors on their conclusions regarding the addition of novel proxies.

Page 26 line 611 –twice the use of the word "first"

Page 27 line 633: The authors name the Denvil-Sommer method as a way forward. I agree that this is an exciting new method, however – in the context of the text lines above – it is also not a fundamentally different method, as it is also based on machine learning.

---

## Editor Comment (EC1) · Andrew Yool (Editor) · 20 May 2019

Over the weekend, a comment was posted that directed viewers to entirely unrelated (and off-topic) research rather than a commentary on the manuscript of Gregor et al. I have arranged for this comment to be deleted, and would encourage visitors and potential commentators to please stick to discussing this manuscript in this forum.

Andrew Yool

2019.

---

## Referee Comment (RC2) · Jamie Shutler (Referee) · 22 May 2019

The authors present a novel intelligent interpolation method and a comprehensive analysis of current methods to intelligently interpolate pCO2 data (which then enable the calculation of global and regional atmosphere-ocean gas fluxes and net sink values). The analysis concludes that we are reaching the limit of improvement in these methods and that a increase in data coverage within those regions sparsely sampled is now needed to allow further advancement.

Whilst I agree with the final conclusion, I can see that the methods in the the analysis would benefit from further refinement to enable the 'wall' to be correctly identified. This refinement should focus on the consistent and more thorough handling of temperature within the data used for training and verifying the pCO2 interpolation methods. Improvements within the air-sea gas flux calculation itself would also enable a more accurate flux calculation.

Hence I would recommend revision of this paper before acceptance. This will require updating the methods and re-running the analysis.

Here are the major points that would need to be addressed. I suspect that addressing these temperature related issues will improve the overall results and conclusions.

The temperature issue relates to the section in the discussion that focuses on uncertainty. The strong temperature dependence of pCO2 means that a consistent handling of temperature is needed throughout the analysis to ensure that (inconsistent handling of ) temperature is not the source of any biases and/or accuracy issues. This is especially true for a large collated dataset which has been generated from data collated from multiple systems, regions and ships (i.e. the SOCAT data). The SOCAT dataset is an amazing resource and vital in the study of air-sea CO2 exchange, but using it directly (in its original form) for this sort of global air-sea gas flux study can lead to unknown errors and biases. The need for correct temperature handling in air-sea gas exchange studies has been reviewed in depth by Woolf et al., (2016). The impacts of incorrect and inconsistent temperature handling can be significant, especially within large collated datasets and global analyses (please see the different examples and impacts that temperature can have on the different components of the gas flux calculation that are detailed within Woolf et al. 2016).

1. The SOCAT gridded dataset are based on gridded data that have all originated from different ships, systems and times. Hence the gridded values are likley to contain unknown biases due to inconsistent depths and thus temperatures that were originally

GMDD
captured/linked to each pCO2/SST pair, but which was lost as a result of gridding (as SST and pCO2 are gridded individually). Furthermore, the differing depths of each sample means that multiple measurements within each box could be from different depths and so they are not part of the same (statistical) population. This issue can be overcome by first re-analysing the original SOCAT cruise data to a common temperature dataset (and thus a common sampling depth) and then re-gridding them (through the use of a satellite observed sea surface temparature dataset). This theory is explained in Woolf et al., 2016, the method is described in Goddij-murphy et al (2015) and the software tool to enable this is avialable within the FluxEngine open source toolbox (Shutler et al., 2016). A link to the github repository is below.

This needs to be done for the data used to train the methods and also data used to verify the performance of the methods.

2. The gas flux calculation itself as used by the authors (equation 2) is likely to add temperature related errors into the analysis. This bulk formulation using DpCO2 ignores vertical temperature gradients and so is likely to introduce additional (and unknown) errors into the analysis. Woolf et al., (2016) also discusses the shortfalls of using an inaccurate gas flux calculation. The work would benefit from using a version of the equation that accounts for differing solubilities at the top and bottom of the mass boundary layer. Section 2 of Woolf et al., (2016) provides the information needed to achieve this.

3. All of the independent methods used for the secondary validation also suffer from the issues of inconsistent temperature handling. This was a shortfall of the Rodenbeck et al. 2015. The work of Rodenbeck et al., 2015 was an excellent first step, but any verification using these data should consider the impact that inconsistent temperature handling is likley to have on any derived results. The authors should recognise this and discuss the issue.

4. The independent data used for verification (e.g. from GLODAPv2) will also suffer
from the same temperature issue. the GLODAPv2 data is also an amazing and very useful dataset. However, the data are all sampled from different depths and using different methods. So any conclusions drawn from deriving PCO2 from these data and its use for verification should consider the impact of inconsistent sampling depths and the resultant (unknown) temperature biases that this will introduce. The authors should recognise this and discuss the issue.

5. the GLODAPv2 data and its accompanying publication (Pfeil et al) provides an estimate of the bias within the GLODAPv2 data, which the authors here have used as an estimate of the uncertainty for the GLODAPv2 data. The bias will only be one component of the uncertainty (especially when using a large hisorical dataset like GLODAPv2 which spans many years, a period over which signifcant advances in methods have been made). The authors should mention and discuss this issue.

6. the GHRSSt dataset that is used within the analysis is (i think) a combined thermal and microwave dataset and so its valid for a specific depth (if mostly microwave data then its most likely sub-skin) making it valid for the bottom of the mass boundary layer (which is key when considering calculating the pCO2 and relevant solubility). if this is used as the input dataset for training the different interpolation methods then it would seem sensible to use this dataset for the gas flux calculation (to represent the temperature at the bottom of the mass boundary layer). This could also be used as the reference for re-analysing the SOCAT data prior to re-gridding. Doing all of this will ensure that your methods are consistent throughout.

Minor point, but still important: 6. Please can the authors clarify their descriptions of satellite observations? Satellite observations are not proxies (as stated on line 670). This is a common misconception, satellite observations are precise and accurate measurements of electromagnetic energy. e.g. some eatellite observations of SST are thermal infrared measurements and can be more accurate and precise than in situ sea surface temperature measurements (O'carroll et al 2008). The difference between satellite and in situ meaurements or observations (apart from the method of collecting
them) is predominantly the depth that the meaurement is valid for, as satellite sensors will retrieve the skin or sub-skin temperature, whereas an in situ measurement (which is a voltage measurement through a thermally sensitive resistor that is then calibrated to temperature) is normally collected at a few metres depth or somewhere near the surface. These differences are also briefly discssued in Woolf et al., 2016. For satellite chl-a its typically a visible spectrum measurement that uses a empirical relationship to calibrate the optical measurement to estimate chl-a. So the sentence (line 670) should be corrected to say 'satellite observations' (rather than proxy). please can the author check that the rest of their manuscript for further instances of the same issue?

References O'carroll et al (2008) Three-Way Error Analysis between AATSR, AMSR-E, and In Situ Sea Surface Temperature Observations, https://journals.ametsoc.org/doi/full/10.1175/2007JTECHO542.1

Woolf DK, Land PE, Shutler JD, Goddijn-Murphy LM, Donlon CJ (2016). On the calculation of air-sea fluxes of CO 2. in the presence of temperature and salinity gradients. Journal of Geophysical Research: Oceans, 121(2), 1229-1248.

Goddijn-Murphy L, Woolf DK, Callaghan AH, Nightingale PD, Shutler JD (2016). A reconciliation of empirical and mechanistic models of the air-sea gas transfer velocity. Journal of Geophysical Research: Oceans, 121(1), 818-835.

Shutler JD, Land PE, Piolle J-F, Woolf DK, Goddijn-Murphy L, Paul F, Girard-Ardhuin F, Chapron B, Donlon CJ (2016). FluxEngine: a Flexible Processing System for Calculating Atmosphere–Ocean Carbon Dioxide Gas Fluxes and Climatologies. Journal of Atmospheric and Oceanic Technology, 33(4), 741-756.

FluxEngine software to perform the re-analysis and an accurate gas flux calculation can be found here: https://github.com/oceanflux-ghg/FluxEngine

---

## Referee Comment (RC3) · Anonymous Referee #3 · 28 May 2019

The authors present (1) a new algorithm to spatiotemporally interpolate discrete pCO2 measurements into continuous pCO2 field, and (2) present and discuss a comparison between this and existing pCO2 interpolations in the light of several metrics. Concerning (1), though the algorithm is based on the same principles (namely non-linear regression of pCO2 against driving quantities measured more completely) which have also been employed by several existing algorithms for the same purpose, it differs by a formalized selection of how to split the ocean into areas of similar biogeochemical behaviour. In particular, the selection is not done in isolation but involves the regression

step itself. This split into coherent areas is an important element in regression-based pCO2 interpolations. It is therefore an interesting contribution worth to be published. The comparison (2) is a useful part of the evaluation. In the discussion, however, the authors somewhat delve in general statements that have already been discussed in the literature or have been tackled in ongoing research projects (see below). I also think that some statements may be put into perspective (see below). The paper is well written, though at a number of places the text may be revised to become more accessible (see some suggestions below). In summary, I'd like to recommend to publish this study in GMD, after revisions to address the points detailed in the following.

On terminology, there is a problem with the authors' use of "ensemble". Usually, an "ensemble" means a set of several members. At various places (first in L124), however, the word is apparently used for "ensemble mean" (= just one entity, not a set any more). This sometimes distorts the meaning and confused me substantially on first reading. Similarly confusing is the use of "clusters" not only for "points belonging together" but also for "cases having different (or differently many) clusters".

As an interesting feature in Fig 7(a), I notice adjacent bands of strong opposite biases in the eastern Pacific. Could this point to an inappropriately located boundary between the regions? It may help to check if these bands also occur for K21E and BIO23 individually, do they? If so, is there a systematic difference in the location of the region boundary between K21E and BIO23? I imagine that such analysis might give hints on how to improve the interpolation.

The paper also discusses more general aspects of pCO2 interpolation, such as the potential "wall" mentioned in the title, which is definitely an interesting and relevant question. However, I'm a bit surprised by some formulations, such as L677-678 or L578 ("stagnant"), which seem to suggest that "there must be intrinsic limits if not even our method performs better than other methods". Why should we expect your particular method to exhaust all what's achievable? After all, the presented method is based, like several previous pCO2 gap-filling studies, on instantaneous relationships to physical or

biological oceanic quantities. While such relationships have proven to capture a good fraction of pCO2 variability, it is clear that oceanic biogeochemistry is a dynamical system, ie., pCO2 depends not only on the current state but on the past history. Though the need of "new methods" is being mentioned (L629-L638), the discussion remains solely with regressions. For example, it ignores that other approaches like "data assimilation" into process models do exist already (though mostly not yet in a stage to fit the data closely). On the other hand, the discussion likewise ignores that sophisticated methods like regressions against drivers or data assimilation are only needed because large data gaps need to be bridged. In a data-rich world, such as pleaded for by the authors, simpler auto-regressive methods are also sufficient, as indicated e.g. in your Fig 10 by the relatively good agreement of driver-regressions and auto-regressions in the more data-covered areas. In order to make the discussion more interesting in the revised paper, therefore, I feel that it should be done in a wider context of the existing literature and make more concrete statements on how to go forward. An alternative option may be to substantially shorten the discussion and keep ideas for a future research paper (why not in the context of a new SOCOM as the authors propose?).

The same also applies to the discussion of sampling strategies. The dependence of accuracy on data density, the need for denser sampling in many parts of the southern hemisphere, or the use of synthetic data to test sampling strategies, are all not new. While autonomous sampling devices are presented as "a new way", there are papers already, e.g., from the SOCCOM project, which are not even cited in the discussion. These papers already discuss possibilities and limits on a higher level. In my opinion, a discussion of sampling also needs the input by the experimentalists (e.g., I'm not sure how "low-cost" the autonomous platforms really are). I feel the paper would win from a shorter and more concise discussion.

The authors find that the average over their ensemble of regressions perfomed better under several metrics than the individual ensemble members. They present this as a contradiction to warnings against the use of ensemble averages in the literature.

When comparing the presented ensemble of pCO2 interpolations with e.g. the SOCOM ensemble, however, there seem to be distinct differences in how statistically homogeneous these ensembles are: The presented ensemble of regressions against the same explanatory variables likely spreads in the finer details (see the rather similar behaviour in Fig 6) such that averaging may reduce noise, while there are large systematic differences (including members with limited ability to fit the data) in the SOCOM ensemble or other ensembles from the literature. Therefore, I feel the authors should discuss better the conditions under which the average over an ensemble really is a meaningful estimate.

Minor comments:

L47: maybe add "e.g."

L48: "maximise" does not seem to be the right word here

L57: "consolidated" probably means "collated" or similar

L65: as far as I see, there is not actually any weighting in that paper

L67: from my knowledge of the literature, most studies analysing existing pCO2 interpolations actually use several of these, rather than one "most widely used method"

L106: maybe add "separately" after "applied"

L107: "K-means clustering" should be briefly explained (either here or later, e.g. around L232)

L115: "described by" may better be "denoted as" (same in L120)

L117: you probably mean "a range of 11 to 25 clusters"

L123: spurious "and"

Fig 1, section "DATA": Table "XX"

Fig 1 Sect 4 (and many other places in text, tables, and figures): "HOTS" should be

[Figure]

"HOT"

L140: maybe add "a" before "predictive"

L155: Explain or spell out "GHRSST"

L172: Is the use of "random noise" a standard technique to fill incomplete input data? Isn't there a chance that this creates instability to the regression? A brief explanation or a reference would be useful here

L188: Maybe add "separate" or "individual" before "regressions"

L198: "palate" is probably misspelled, what about "shown by separate colors"

L214: Reference to Fig 5 would break the order of figures, but could easily be removed here

L216: not sure "a.k.a." is a suitable abbreviation

L217: "80:20" seems to contradict "75:25" in Fig 1

L224-235: I found this paragraph difficult to understand. Can you say more exlicitly which "hyper-parameters" you mean? It may also help to better link this paragraph and the previous one.

L244: add "below" after "Eqn 3 and 4"

L247-257: I did not find this paragraph very clear. Does it mean that you repeat the previous steps with other selections of years in the test-training split?

L300: The RˆIAV metric was first mentioned in the SOCOM paper (ie. Rödenbeck et al., 2015)

L303-304: According to Rödenbeck et al. (2015), their benchmark has no interannual variability, but it does have seasonal variability.

L306-307: Also here, the metric used by the authors (or at least the description of it) is

not the same as that presented in Rödenbeck et al. (2015): SOCOM used the standard deviation over yearly averaged pCO2 mismatches, not standard deviations over the full data in individual years. If the metric used in this study has indeed been calculated in the way it is described, it should be sensitive to within-year variations (probably dominated by the match to the seasonal cycle) but be insensitive to interannual variations.

L309-310: A benchmark constructed this way still contains the interannual variations of pCO2 (as the atmospheric pCO2 has very little IAV compared to seawater pCO2, except for the rising trend). Also here, this is opposite to what has been described by Rödenbeck et al. (2015) which removed IAV from the benchmark.

L313: I guess you mean "in contrast to the standard deviation which is sensitive to outliers."

L314: It is not clear to me what "interannual resampling" means, please be more explicit.

L319: What do you mean by "second part of the experiment"? Is it the "regression step of the algorithm"?

Fig 5: Why can e.g. D ever be worse than C, given that D has more degrees of freedom than C?

L326-327: If Fig 5 is showing the averages of the scores from the 4 methods, I was wondering whether these 4 methods show the same general behaviour (ie. whether the better/worse scores occur in similar rows/columns)? I'm asking because only if yes, Fig 5 would give a meaningful picture about which number of clusters and which features are best. A statement on that should be added.

L343: This has been said before and should be omitted here.

L344: Contradictory use of the term "ensemble", see remark above

L347: add "average" after "ensemble"

Tab 3 (also Tab4): I think the 1st column should better termed "clustering"

L355: I guess you mean "for each number of clusters"?

L366: I was wondering whether the occurrence of lower biases in the test years may actually be systematic (ie. not by chance). Are the same years (as listed in Sect 2.4) used in all the regressions? If so, couldn't it be that they implicitly lead to low biases through the model selection?

L380: You probably mean "sampling density"?

L397 and 399: The "||" around the unit is a rather sloppy notation. Better be explicit by writing "|bias| < 5uatm, RSME < 10uatm".

L410: Duplicate "bias"

L411: Add comma after "ERT", otherwise difficult to read.

L426: Are the statistics shown in the Taylor diagrams calculated over all individual data points? That is, do they reflect both spatial and temporal features?

Fig 8: Consider to use the same radial axis limits for all 6 Taylor diagrams. For example, the estimates seem to lie more apart for HOT, but that's only because the variability at HOT is smaller than in others of the independent data sets.

L436: Spurious "that"

L469: "2002" seems to contradict "2000" in L475 and 480.

L470: Shouldn't "regions" be "region groups"?

L479: It seems to me that "reflect" would better fit than "highlight"

L523-525: You could nicely link this back to Fig 2

L526: Add "trend" after "NH-ST", as it is mainly the trend which is reflected by IQRˆIA (if I understood correctly)

L544: replace "ensembles" by "ensemble averages" (see remark on terminology above)

L593, Fig 12: It remains unclear to me how to interpret the "seasonal cycle reproducibility". Doesn't it get smaller with stronger IAV? A short explanation would be helpful.

L611: duplicate "first"

L628: According to the paper, the resolution is daily, not 6-hourly.

L623: What do you mean by "procedural architectures"?

L633: The method by Denvil-Sommer et al. (2018) is named as an example of a "fundamentally new method". In fact, however, the "CARBONES-NN" contribution to the SOCOM ensemble also employed a climatological and an interannual step, but did not outperform other methods there. To clarify this interesting question, I'd suggest to include the Denvil-Sommer et al. (2018) results into your comparison Sect 3.3-3.5, as this would allow a clean comparison.

L645: If I understood correctly, the IQR^IA metric is specifically sensitive to the trend. Why do you particularly propose this metric in the sampling context?

L695: Is "spatial coherence" really the right word here?

---

## Author Comment (AC1) · 28 Jun 2019

**Authors Comments**

We would like to thank the reviewers for their thoughtful and constructive feedback, which we think will strengthen the paper and its ideas. Note that we only produce our comments to the reviewers' comments here and do not show the actual changes made to the manuscript. We include some of the reviewers' concerns and statements which are shown in *blue* to distinguish them from our text and suggested corrections.

**Response to R1 (Peter Lanschützer)**

R1's comments were minor and no major changes were suggested. We will implement all the suggestions made by the reviewer. The reviewer made one point which we feel is important to address here:

*Page 24 and onward: The authors argue for the inclusion of EKE to move forward, but I was not fully convinced, given the small improvement in Figure 5. Another limiting factor that is not discussed is availability. In an ideal case, one would use time-varying fields of SST, SSS, DIC, TALK, etc. however, in most cases they are not available. As the authors mention, EKE only exists as a climatology, hence one cannot expect directly improved IAV signals from it (as e.g. visible in Figure 5c). I do however agree with the authors on their conclusions regarding the addition of novel proxies.*

An important point to make is that we use EKE only as a clustering variable. It will thus only impact pCO2 estimates through the impact that EKE has on clustering. In other words, EKE could capture regions where pCO2 might respond differently to drivers. However, the reviewer is correct in saying that it will not improve the IAV as we use a climatology of EKE.

**Response to R2 (Jamie Shutler)**

R2 had several suggestions in ways to improve the manuscript and were themed primarily around the way in which we deal with SST in our study. The reviewer makes three major suggestions: 1) temperature correction to the pCO2 based on the difference between SOCAT temperatures (ship based) and the optimally interpolated SST; 2) correct handling of SST in the calculation of air-sea $CO_2$ fluxes; 3) both machine learning methods and validation data suffer from the same temperature uncertainties as highlighted in points 1 and 2. An important point to note is that we have in fact used the optimally interpolated SST by Reynolds et al. (2007) which is estimated only from AVHRR data.

1. *The SOCAT gridded dataset is based on gridded data that have all originated from different ships, systems and times. Hence the gridded values are likely to contain unknown biases due to inconsistent depths and thus temperatures that were originally captured/linked to each pCO2/SST pair, but which was lost as a result of gridding (as SST and pCO2 are gridded individually). Furthermore, the differing depths of each sample means that multiple measurements within each box could be from different depths and so they are not part of the same (statistical) population. This issue can be overcome by first re-analysing the original SOCAT cruise data to a common temperature dataset (and thus a common sampling depth) and then re-gridding them (through the use of a satellite-observed sea surface temperature dataset).*

We find the reviewer's first point interesting and potentially important — it is something that we had not considered. We have made a preliminary investigation on this discrepancy between SOCAT SST and optimally interpolated AVHRR SST and the impact it has on $p$CO$_2$.

We used the raw (and not gridded) SOCAT data for this experiment. This discrepancy is surprisingly large with the 25th and 75th percentiles being ~ -18 and ~14 µatm. However, the median bias of $p$CO$_2$ is -1.59 µatm and the error is pseudo-normally distributed. The spatial distribution of these "biases" are shown on the map on the following page. There seems to be little to no spatial pattern (latitudinal or basin) in the data (at least at our superficial level of investigation). However, in some cases it seems that the along-track decorrelation length scale is longer than completely random noise would suggest, indicating that biases may be driven by the structure/stratification of the upper layer of the water column. There are other factors that could also contribute to these biases such as ship specific biases (warming of intake water before thermosalinograph).

[Figure]

(left) A histogram of temperature residuals (NCEI - SOCAT);

(right) A histogram of fCO2 residuals when fCO2 is corrected (NCEI - SOCAT)

[Figure]

Map of fCO2 difference between SOCAT SST and AVHRR SST (µatm) using correction as suggested by the reviewers: $pCO_2^{AVHRR} = pCO_2^{SOCAT} \times e^{(0.0423 * \Delta T)}$. Note that all SOCAT flags were included.

We performed a second preliminary investigation of the impact that correcting pCO2 would have on the machine learning estimates. The experimental setup for this experiment is somewhat simplified (for the sake of time). We use only gradient boosting and the same testing years used in the manuscript training. We also do not use clusters for this first order approach. We perform three regressions to compare:

1) a control where SOCAT SST is used without correcting $pCO_2$;

2) OISST AVHRR temperature is used but we do correct $pCO_2$ for temperature;

3) OISST AVHRR temperature is used and we correct $pCO_2$ for temperature.

We show these results in the table below

| Test results based on gridded 1° SOCAT data | MAE (µatm) | RMSE (µatm) | $r^2$ |
|---|---|---|---|
| **SOCAT temperatures no pCO$_2$ correction** | 12.93 | 20.65 | 0.73 |
| **AVHRR temperatures no pCO$_2$ correction** | 13.35 | 21.07 | 0.72 |
| **AVHRR temperatures pCO$_2$ corrected to AVHRR** | 14.42 | 22.74 | 0.68 |

The results show relatively large RMSE values compared to the results in the experiment and this is likely because data is not clustered. The magnitude of these results relative to each other is the important part here. The SOCAT SST regression scores the lowest errors, but surprisingly, the OISST corrected $pCO_2$ does not perform as well as the uncorrected $pCO_2$.

This result complicates whether or not to implement the corrections suggested by R2. We thus suggest that we will discuss the importance of the way that temperature is handled in machine learning applications of $pCO_2$ – this is part of the problems (the wall) we face. However, we will also include the results from a more complete analysis of the effect of correcting $pCO_2$ to the temperature discrepancy in the supplementary materials.

*2. The gas flux calculation itself as used by the authors (equation 2) is likely to add temperature related errors into the analysis. This bulk formulation using DpCO2 ignores vertical temperature gradients and so is likely to introduce additional (and unknown) errors into the analysis. Woolf et al., (2016) also discuss the shortfalls of using an inaccurate gas flux calculation. The work would benefit from using a version of the equation that accounts for differing solubilities at the top and bottom of the mass boundary layer. Section 2 of Woolf et al., (2016) provides the information needed to achieve this.*

We will use the suggested flux calculation suggested by the reviewer. We will also do this for the comparison datasets (SOMFFN, MLS, MLR) as $pCO_2$ for each product is available and the comparison will only be fair if the same procedure is applied to calculate fluxes.

*3. Both machine learning methods and validation data suffer from the same temperature uncertainties as highlighted in points 1 and 2.*

Based on the results above, the issue of temperature corrections in SOCAT and the validation datasets are clearly an important but complex matter and will thus be discussed.

Other minor concerns by the reviewer will be addressed and corrected in the manuscript.

**Response to reviewer 3 (anonymous)**

R3 provided a very thorough and well thought out critique of the manuscript. The reviewer had several concerns about the manuscript, particularly with the discussion. The reviewer tentatively suggested separating some of the ideas in the discussion into a second paper. We would like to keep the manuscript as one and we hope that the suggestions that we make below will address their major concerns sufficiently.

The first point is that we perhaps claim a little too strongly that "*we have hit the wall*" and by making this claim we assert our model as an exhaustive approach without regard for methods beyond regressive approaches:

> *The paper also discusses more general aspects of pCO2 interpolation, such as the potential "wall" mentioned in the title, which is definitely an interesting and relevant question. However, I'm a bit surprised by some formulations, such as L677-678 or L578 ("stagnant"), which seem to suggest that "there must be intrinsic limits if not even our method performs better than other methods". Why should we expect your particular method to exhaust all that's achievable?*

We will address the reviewer's concern by rephrasing some of the bold statements (e.g. "stagnant") and clarifying the scope and framework of the assessment where we compare only within the statistical gap-filling framework. At the same time, we will broaden the discussion with reference to projects such as the Southern Ocean State Estimate (SOSE) which uses the assimilation of observations to correct toward the truth (Verdy and Mazloff, 2017). Further, we will make it clear that this study strengthens the case for models such as SOSE as there is seemingly (from our work) a limit to what regressive models can achieve. Where the latter point is made quite clear by the Taylor diagrams in which regression methods are tightly grouped for each of the respective validation datasets.

The second major point is that the discussion should be simplified, particularly around the subsection "scale-sensitive sampling strategies":

> *In order to make the discussion more interesting in the revised paper, therefore, I feel that it should be done in a wider context of the existing literature and make more concrete statements on how to go forward … While autonomous sampling devices are presented as "a new way", there are papers already, e.g., from the SOCCOM project, which are not even cited in the discussion. These papers already discuss possibilities and limits on a higher level.*

We agree that the discussion should be shorter. Our suggestion of "scale-sensitive sampling" is perhaps a little premature and is thus not constructive in "inching over the wall". Particularly since this has not yet been tested at even the regional scale. This will thus be removed as the primary focus of the section. Further, we recognise, as the reviewer points out, that major strides are being made by the observational community (e.g. SOCCOM) and the discussion will emphasise this more (e.g. Gray et al. 2018; Bushinsky et al. 2019). Moreover, the successful inclusion

of these data in machine learning estimates of $p$CO$_2$ should be a priority at this stage; in other words, regressive or other models that can incorporate unbalanced data (i.e. new winter data in the Southern Ocean at the end of the time-series) should be explored.

We have removed the quote from the Rödenbeck et al (2015) study from the manuscript as the point made below was also made by R1.

When comparing the presented ensemble of pCO2 interpolations with e.g. the SOCOM ensemble, however, there seem to be distinct differences in how statistically homogeneous these ensembles are: The presented ensemble of regressions against the same explanatory variables likely spreads in the finer details (see the rather similar behaviour in Fig 6) such that averaging may reduce noise, while there are large systematic differences (including members with limited ability to fit the data) in the SOCOM ensemble or other ensembles from the literature.

Removed (L544-546): "*We also discourage any ensemble averaging (or medians, etc.) of full spatiotemporal fields or time series, as this would result in variations that are not self-consistent any more and fit the data less well than individual products.*"

There were many minor corrections that the reviewer identified that will be corrected as required.

Perhaps important to mention from the reviewer's minor comments is that we described the RIAV incorrectly and the IQR^IA unclearly; this will be cleared up. The RIAV was in fact calculated in the same was as Rödenbeck et al. (2015).

**Corrections to be made, but not requested by the reviewers**

- Figure 10: the key for JMA-MLR and Jena-MLS were incorrectly swapped in the submitted manuscript. This will be corrected.
- Correction of SST product: we used the AVHRR product by Reynolds et al (2007) and not the OSTIA product by Donlon et al (2012). This is discussed in more detail in Reviewer 2's comments.
- Data availability: we will make the data available on OCADS (https://www.nodc.noaa.gov/ocads/oceans/) instead of figshare.

---

## Author Response (AR1)

**Authors Comments**

We would like to thank the reviewers for their thoughtful and constructive feedback. We think that their feedback has contributed to strengthening the paper and its ideas.

Reviewers' comments are in black (Line numbers refer to first submission manuscript)

Responses are in blue (Line numbers refer to updated line numbers in the track changes document)

**Summary of large changes**

We have made large changes primarily to the discussion as requested by Reviewer 3. We have also added a section to the supplementary material to address the point made by Reviewer 2 (Jamie Shutler) about the mismatched temperatures between SOCAT and the satellite SST product. We have also opted to use the gas transfer velocity of Nightingale et al. (2000; Ni00) instead of Wanninkhof et al. (2014) as recommended by Goddijn-Murphy et al. (2016), where the former (Ni00) parameterises the effect of bubbles relatively well compared to other methods.

**Corrections made, but not requested by the reviewers**

- Figure 10: the key for JMA-MLR and Jena-MLS were incorrectly swapped in the submitted manuscript. This was corrected.
- Correction of SST product: we used the AVHRR product by Reynolds et al (2007) and not the OSTIA product by Donlon et al (2012). This is discussed in more detail in Reviewer 2's comments.
- Data availability: we will make the data available on OCADS (https://www.nodc.noaa.gov/ocads/oceans/) instead of figshare.

**Reviewer 1: Peter Landschützer**

Gregor and colleagues present an impressive and comprehensive study, comparing the performance of various machine learning-based regression approaches in combi- nation with different ocean-biome combinations. The authors compare their new esti- mates with the current "state-of-the-art" methods represented in the SOCOM intercom- parison project and a wide range of independent and novel validation data. Using this impressive set of data, the authors ask the question, whether we have "hit the wall" in our accuracy to reconstruct the ocean carbon sink, and in what way we may further improve in the future.

Strengths: I am particularly impressed by the amount of data and experiments used by the au- thors. And despite this vast amount of information, the manuscript is clearly written and easy to follow. The study provides a step forward compared to other existing in- tercomparison studies (e.g. the cited papers of Rödenbeck et al 2015 and Ritter et al 2017) as it compares a more consistent set (or ensemble) of estimates, all created within the same regions and with the same observational dataset (SOCATv5). Ad- ditionally, I am not aware of any other study that makes use of such a large set of independent estimates (GLODAPv2, SOCCOM, etc.) to validate their results. All of the above are significant steps forward and provide a well-suited set-up for answering the research questions posed by the authors.

Weaknesses: I have not encountered any major weakness.

Recommendation: This study is a significant contribution to our scientific understanding of current "state-of-the-art" observation-based pCO2 and air-sea CO2 flux estimates, their limitations in space and time as well as potential pathways for future improvement. The study will instantly be of interest and benefit ongoing carbon cycle assessment studies, such

Page 2 line 61: "did not identify" – I would rather say that SOCOM "did not look for" the best method. Unlike this study, the SOCOM comparison was slightly more difficult as some methods were still based on older observational datasets (e.g. LDEOv1) which made an objective comparison with one dataset (e.g. SOCATv5) additionally challenging.

L63: Changed to: *did not seek to identify*

Page 3 line 72-73: Correct. Very little agreement was found in the SO, however, it is worth noting that Ritter et al also found remarkable agreement regarding decadal signals, despite the strong discrepancies in seasonality and IAV.

L76: Added the following text: *Similarly, Ritter et al. (2017) found little agreement in the Southern Ocean on seasonal timescales, yet on decadal time-scales, there was an agreement on the direction of trends between gap-filling methods.*

Page 9 lines 186-187: I find myself arguing a lot against the direct inclusion of spatial coordinates. The reason is that CO2 is not directly affected by longitude or latitude, but rather direct environmental proxies that vary along space and time (e.g. SST). Adding Lat and Lon might decrease the error metric as it replaces an unknown.

However, in a process-sense, it makes much more sense to apply some special selection by regimes, biomes or clusters.

L203: We have added the bold text to make our statement clearer: ***This is due to the fact that pCO2 may respond inconsistently to observable feature-variables in different regions as it is not possible to observe all feature-variables that drive pCO2.*** *A common practice* **to avoid the inclusion of coordinates** *is to separate*  *the ocean into regions where processes that drive pCO2 are coherent and then apply regressions to each region – five of the eight regression methods in Rödenbeck et al. (2015) apply this approach. We adopt two* **such** *approaches to develop regions of internal coherence in respect of CO2 variability, namely regions defined by biogeochemical properties and clusters defined by a clustering algorithm. .*

Page 9 figure 3: I absolutely understand the advantage of adding additional regions to the "blank" Fay and McKinley biomes, but a bit more motivation would be good on how these additional regions were chosen (e.g. it is immediately obvious to combine the EBUS regions, but why not the very small Sea of Japan with the surrounding waters?)

L213: The following was added to the text: *We maintain these as separate regions from the original Fay and McKinley (2015). Their study originally did not classify these regions in the core biomes because the physical and biogeochemical properties were not accounted for by the set thresholds from their study. This would suggest that drivers of CO2 in these regions could be quite different from the adjacent open ocean biomes.*

Page 14 lines 322-323: This is a repeat and can be removed

L361-L364: Removed as suggested

Page 14 Figure 5 and following text: The authors in the text often use words like "out-perform" (e.g. line 355, line 384) or "much larger" (e.g. line 364). As a reader when I hear outperform, I immediately think of a 50% error reduction or similar, whereas in fact not a single bias value in Figure 5 or table 3 and table 4 are above 1µatm. Just to put this into context: The current flag A measurement uncertainty is in the range of 1-2µatm. Hence these differences are small. They might be significant, but certainly don't deserve wording like "outperformed"

Changed this as suggested by adding *slightly* or *marginally*

Page 18 line 410: word "bias" occurs twice – remove first occurrence

460: Removed as suggested

Page 24 lines 544-546: One has to be careful quoting the SOCOM study here. The SOCOM study faced 2 additional differences which made a combination trickier. Firstly, not all methods were based on the same observational dataset (some were built on 1 million data, others on almost 15 million at the time). Secondly, not all methods did span the same geographical extend (with some covering more of the coast, less of the Arctic,

etc.). Based on all these differences (and others), Rödenbeck et al 2015 avoided to provide an ensemble mean estimate.

L596: We removed this quote and associated text to uncomplicate the paragraph.

Page 24 and onward: The authors argue for the inclusion of EKE to move forward, but I was not fully convinced, given the small improvement in Figure 5. Another limiting factor that is not discussed is availability. In an ideal case one would use time varying fields of SST, SSS, DIC, TALK, etc. however, in most cases they are not available. As the authors mention, EKE only exists as a climatology, hence one cannot expect directly improved IAV signals from it (as e.g. visible in Figure 5c). I do however agree with the authors on their conclusions regarding the addition of novel proxies.

L611: We use EKE only as a clustering variable – this means that it will not influence the variance of the clusters, but it can capture regions where pCO2 might respond differently to drivers. The reviewer is right in saying that it will not improve the IAV, but it could improve the fit (RMSE). We have further clarified our choice in using the clustering method including EKE (column E in Figure 5) for 21 clusters in **L374**: *"While the individual regression methods' bias and RMSE scores (Figures S5 and S6 respectively) do not match the distributions exactly, the two selected clustering configurations (black boxes in Figure 5) score consistently low for both metrics (with the exception of ERT – discussed in greater detail further on). We motivate to select only one clustering configuration for the sake of simplicity. Furthermore, we select the configuration with 21 clusters (rather than 23) as fewer clusters further reduces the possible complexity at little cost.  The selected clustering configuration with 21 clusters has the following features: SST, log10(MLDclim), pCO2clim, log10(Chl-aclim), and log10(EKEclim); and is hereinafter abbreviated as K21E (see Figure S2 for the distribution of the climatology for these clusters).*

Page 26 line 611 –twice the use of the word "first"
L665: Removed as suggested

Page 27 line 633: The authors name the Denvil-Sommer method as a way forward. I agree that this is an exciting new method, however – in the context of the text lines above – it is also not a fundamentally different method, as it is also based on machine learning.

L694-L704: This section has been removed as large changes have been made to the discussion.

**Reviewer 2: Jamie Shutler**

The authors present a novel intelligent interpolation method and a comprehensive analysis of current methods to intelligently interpolate pCO2 data (which then enable the calculation of global and regional atmosphere-ocean gas fluxes and net sink values). The analysis concludes that we are reaching the limit of improvement in these methods and that an increase in data coverage within those regions sparsely sampled is now needed to allow further advancement.

Whilst I agree with the final conclusion, I can see that the methods in the analysis would benefit from further refinement to enable the 'wall' to be correctly identified. This refinement should focus on the consistent and more thorough handling of temperature within the data used for training and verifying the pCO2 interpolation methods. Improvements within the air-sea gas flux calculation itself would also enable a more accurate flux calculation. Hence I would recommend revision of this paper before acceptance. This will require updating the methods and re-running the analysis.

Here are the major points that would need to be addressed. I suspect that addressing these temperature related issues will improve the overall results and conclusions.

The temperature issue relates to the section in the discussion that focuses on uncertainty. The strong temperature dependence of pCO2 means that a consistent handling of temperature is needed throughout the analysis to ensure that (inconsistent handling of) temperature is not the source of any biases and/or accuracy issues. This is especially true for a large collated dataset which has been generated from data collated from multiple systems, regions and ships (i.e. the SOCAT data). The SOCAT dataset is an amazing resource and vital in the study of air-sea CO2 exchange, but using it directly (in its original form) for this sort of global air-sea gas flux study can lead to unknown errors and biases. The need for correct temperature handling in air-sea gas exchange studies has been reviewed in depth by Woolf et al., (2016). The impacts of incorrect and inconsistent temperature handling can be significant, especially within large collated datasets and global analyses (please see the different examples and impacts that temperature can have on the different components of the gas flux calculation that are detailed within Woolf et al. 2016).

We tackle this point in details below, but would like to highlight beforehand that we made a mistake in the citation of the SST dataset used - we originally cited the OSTIA product by Donlon et al. (2012), while it was in fact the dOISSTv2 product that uses only AVHRR data (Reynolds et al. 2007; Banzon et al. 2016), a product that reports the bulk temperature (between 0.5 and 1.0 m depth).

1. The SOCAT gridded dataset are based on gridded data that have all originated from different ships, systems and times. Hence the gridded values are likely to contain unknown biases due to inconsistent depths and thus temperatures that were originally captured/linked to each pCO2/SST pair, but which was lost as a result of gridding (as SST and pCO2 are gridded individually). Furthermore, the differing depths of each sample means that multiple measurements within each box could be from different depths and so they are not part of

the same (statistical) population. This issue can be overcome by first re-analysing the original SOCAT cruise data to a common temperature dataset (and thus a common sampling depth) and then re-gridding them (through the use of a satellite observed sea surface temperature dataset). This theory is explained in Woolf et al., 2016, the method is described in Goddij-murphy et al (2015) and a software tool to enable this is available within the FluxEngine open source toolbox (Shutler et al., 2016). A link to the github repository is below. This needs to be done for the data used to train the methods and data used to verify the performance of the methods.

This is a very interesting point, and one that we have not considered. We have done an experiment in the supplementary material (S2.4) that investigates this issue. This is done in two parts: a) difference between SOCAT $pCO_2$ calculated from ship intake temperature and skin temperature as measured by satellite; b) the impact that this has on machine learning estimates of $pCO_2$.

We show the following figure in the SM (as Figure S3). According to Banzon et al. (2016) who released the dOISSTv2 product, the global positive temperature bias (0.13°C, panel (a) below) is due to warming in the engine room between the water intake and the thermosalinograph. With this bias removed, the distribution is heterogeneous on a large scale, but there still seem to be regional biases (e.g. overall negative bias in the high latitude and positive bias in the low latitudes).

[Figure]

In part two of the experiment we show that the temperature adjustment actually results in larger RMSE scores and stronger biases (table below). Moreover, the regionality in the biases still persists. We have thus opted not to apply this correction.

|  | Bias | MAE | RMSE | r² |
|---|---|---|---|---|
| (a) SOCAT SST / no $pCO_2$ correction | -0.23 | 12.15 | 18.83 | 0.74 |
| (b) OISST / no $pCO_2$ correction | 0.00 | 12.43 | 19.17 | 0.73 |
| (c) OISST / $pCO_2$ corrected to OISST | -0.50 | 13.55 | 20.94 | 0.7 |

*To investigate these errors further we plotted the distribution of the biases as shown in Figure S4. The distribution for the biases is very similar for all experiments. This illustrates that the models' capability*

*to reproduce the observations is a far greater contributor to the error than the error attributed to the temperature discrepancy.*

[Figure]

**Figure S4:** *The biases from three regression experiments testing the effect of correcting $pCO_2$ to the temperature discrepancy between SOCAT temperatures and the OISST product. The figure numbers correspond to the first column in Table S1. Note that the spatial biases are robust to the temperature corrections.*

**L163-L171:** We have added the following text to the main manuscript to address this: *A consideration in this application of SOCAT data is the mismatch between the temperature at which pCO2 is measured ($SST^{SOCAT}$) and the OISST (Goddijn-Murphy et al. 2015). The OISST product reports the bulk temperature at ~1 m, whereas $SST^{SOCAT}$ is measured at the depth of a ship's water intake (Banzon et al., 2016; Bakker et al. 2016). A comparison of $SST^{SOCAT}$ and OISST shows that the former has a warm global mean bias of 0.13°C (Figure S3a), the same as that reported by Banzon et al. (2016), which they attribute to warming in the ships intake. Further investigation shows that correcting pCO2 for the temperature bias reduces the accuracy of the machine learning estimates relative to the training data (Section S2.4) and does not improve spatial biases. We thus do not apply the correction applied in Goddijn-Murphy et al. (2015).*

We further address this in the discussion (**L683 - L693**), urging the community to explore these potentially important issues in a coordinated way. The impact of temperature biases are not negligible with the first quartile of the ΔpCO2 differences being ~ -7 μatm.

2. The gas flux calculation itself as used by the authors (equation 2) is likely to add temperature related errors into the analysis. This bulk formulation using DpCO2 ignores vertical temperature gradients and so is likely

to introduce additional (and unknown) errors into the analysis. Woolf et al., (2016) also discusses the shortfalls of using an inaccurate gas flux calculation. The work would benefit from using a version of the equation that accounts for differing solubilities at the top and bottom of the mass boundary layer. Section 2 of Woolf et al., (2016) provides the information needed to achieve this.

**L323-328:** We have considered this correction extensively, reading through Woolf et al. (2016) and even the FluxEnging code, and applied it to the CSIR-ML6 flux estimates to test this difference. While we feel that this is an important consideration in the calculation of sea-air CO2 fluxes, we think that it does not change the message of this publication which is focussed primarily on pCO2 and not fluxes. Given that we do not have immediate access to the temperature inputs of the comparison datasets (JENA, MPL, UEA, JMA), applying this correction only to CSIR-ML6 is not consistent. We thus stick with the bulk fluxes approach. We have however included the variables required to calculate fluxes using the RAPID model ($K_0^{skin}$, $fCO_2^{skin}$, $K_0^{fnd}$, $fCO_2^{fnd}$) in the final netCDF file that will be shared on OCADS, where *skin* is an approximation of the interfacial layer and we make the assumption that the base temperature is equivalent to the foundation (*fnd*) temperature (Woolf et al. 2016).

3.  All of the independent methods used for the secondary validation also suffer from the issues of inconsistent temperature handling. This was a shortfall of the Rodenbeck et al. 2015. The work of Rodenbeck et al., 2015 was an excellent first step, but any verification using these data should consider the impact that inconsistent temperature handling is likely to have on any derived results. The authors should recognise this and discuss the issue.

    **L475:** We have added the following text: *Note that these datasets (with the exception of the UEA-SI) will also suffer from the same temperature biases discussed in S2.4.*

4.  The independent data used for verification (e.g. from GLODAPv2) will also suffer from the same temperature issue. the GLODAPv2 data is also an amazing and very useful dataset. However, the data are all sampled from different depths and using different methods. So any conclusions drawn from deriving PCO2 from these data and its use for verification should consider the impact of inconsistent sampling depths and the resultant (unknown) temperature biases that this will introduce. The authors should recognise this and discuss the issue.

    **L310-314:** We have picked this up in the discussion and made reference to the fact that these validation datasets likely also suffer from temperature mismatches. However, we also add that because we deal with each validation dataset independently, this issue does not impact the outcome of the assessment.

5.  The GLODAPv2 data and its accompanying publication (Pfeil et al) provides an estimate of the bias within the GLODAPv2 data, which the authors here have used as an estimate of the uncertainty for the GLODAPv2 data. The bias will only be one component of the uncertainty (especially when using a large historical dataset like GLODAPv2 which spans many years, a period over which significant advances in methods have been made). The authors should mention and discuss this issue.

**L305:** The following text was added to address this point: *Additionally, GLODAP v2 data has been adjusted on a per-profile bases to minimise the biases through the comparison of deep slow-changing ocean properties (Olsen et al. 2016).*

225

6.  The GHRSSt dataset that is used within the analysis is (i think) a combined thermal and microwave dataset and so its valid for a specific depth (if mostly microwave data then its most likely sub-skin) making it valid for the bottom of the mass boundary layer (which is key when considering calculating the pCO2 and relevant solubility). if this is used as the input dataset for training the different interpolation methods then it would

230     seem sensible to use this dataset for the gas flux calculation (to represent the temperature at the bottom of the mass boundary layer). This could also be used as the reference for re-analysing the SOCAT data prior to re-gridding. Doing all of this will ensure that your methods are consistent throughout.
        The temperature data was incorrectly cited, where we cited Donlon et al. (2012) the data used is the updated dOISSTv2 product (an update to Reynolds et al. 2007 which is referenced as Banzon et al. 2016). This

235     temperature product reports bulk SST (between 0.5 and 1 m). We thus have to make the assumption that the bulk temperature is representative of foundation temperature. However, as mentioned, we do not implement the RAPID model in flux calculations for the paper, but we do provide the components required to calculate pCO2 as defined in Woolf et al. (2016). Please see our response to point 2 for further details.

7.  Minor point, but still important: Please can the authors clarify their descriptions of satellite observations?
240     Satellite observations are not proxies (as stated on line 670). This is a common misconception, satellite observations are precise and accurate measurements of electromagnetic energy. e.g. some satellite observations of SST are thermal infrared measurements and can be more accurate and precise than in situ sea surface temperature measurements (O'carroll et al 2008). The difference between satellite and in situ measurements or observations (apart from the method of collecting them) is predominantly the depth that the

245     measurement is valid for, as satellite sensors will retrieve the skin or sub-skin temperature, whereas an in situ measurement (which is a voltage measurement through a thermally sensitive resistor that is then calibrated to temperature) is normally collected at a few metres depth or somewhere near the surface. These differences are also briefly discussed in Woolf et al., 2016. For satellite chl-a its typically a visible spectrum measurement that uses a empirical relationship to calibrate the optical measurement to estimate chl-a. So the

250     sentence (line 670) should be corrected to say 'satellite observations' (rather than proxy). please can the author check that the rest of their manuscript for further instances of the same issue?
        The text on L70 has been removed to simplify the discussion as requested by Reviewer 3. However, we feel that this is sufficiently clear throughout the rest of the manuscript, where proxy variables refer to variables that may impact pCO2 and are thus used as predictors.

**Reviewer 3: Anon**

The authors present (1) a new algorithm to spatiotemporally interpolate discrete pCO2 measurements into continuous pCO2 field, and (2) present and discuss a comparison between this and existing pCO2 interpolations in the light of several metrics. Concerning (1), though the algorithm is based on the same principles (namely non-linear regression of pCO2 against driving quantities measured more completely) which have also been employed by several existing algorithms for the same purpose, it differs by a formalized selection of how to split the ocean into areas of similar biogeochemical behaviour. In particular, the selection is not done in isolation but involves the regression step itself. This split into coherent areas is an important element in regression-based pCO2 interpolations. It is therefore an interesting contribution worth to be published. The comparison (2) is a useful part of the evaluation. In the discussion, however, the authors somewhat delve in general statements that have already been discussed in the literature or have been tackled in ongoing research projects (see below). I also think that some statements may be put into perspective (see below). The paper is well written, though at a number of places the text may be revised to become more accessible (see some suggestions below). In summary, I'd like to recommend to publish this study in GMD, after revisions to address the points detailed in the following.

On terminology, there is a problem with the authors' use of "ensemble". Usually, an "ensemble" means a set of several members. At various places (first in L124), however, the word is apparently used for "ensemble mean" (= just one entity, not a set any more). This sometimes distorts the meaning and confused me substantially on first reading. Similarly confusing is the use of "clusters" not only for "points belonging together" but also for "cases having different (or differently many) clusters".

The reviewer raises a valid point about the ambiguity in the use of "ensemble" and "cluster". We have appropriately changed the use of ensemble to ensemble mean/average throughout the manuscript (this is shown in the document with the track changes). Where appropriate, cluster was changed to clustering configuration.

As an interesting feature in Fig 7(a), I notice adjacent bands of strong opposite biases in the eastern Pacific. Could this point to an inappropriately located boundary between the regions? It may help to check if these bands also occur for K21E and BIO23 individually, do they? If so, is there a systematic difference in the location of the region boundary between K21E and BIO23? I imagine that such analysis might give hints on how to improve the interpolation.

Yes, agreed! We highlight these adjacent biases in the discussion in greater detail and use this as motivation for the development of methods that are able to resolve these juxtaposed biases (4.3.1 Reducing existing biases on **L667**).

The paper also discusses more general aspects of pCO2 interpolation, such as the potential "wall" mentioned in the title, which is definitely an interesting and relevant question. However, I'm a bit surprised by some formulations, such as L677-678 or L578 ("stagnant"), which seem to suggest that "there must be intrinsic limits if not even our method performs better than other methods". Why should we expect your particular method to

290   exhaust all what's achievable? After all, the presented method is based, like several previous pCO2 gap-filling studies, on instantaneous relationships to physical or biological oceanic quantities.

We recognise that "*relatively stagnant*" is perhaps slightly overcritical and suggest that using "***plateaued***" may perhaps better. However, we do not make the assumption that we exhaust all options as we suggest that there is a way forward (and we also recognize that our suggestions are not all encompassing). But, as so many studies

295   have pointed out (Rödenbeck et al. 2015; Landschützer et al. 2014; Ritter et al. 2017), that the available data is a big limitation to improved estimates. This has been added on **L713-L739 (Section 4.3.2)**.

While such relationships have proven to capture a good fraction of pCO2 variability, it is clear that oceanic biogeochemistry is a dynamical system, ie., pCO2, depends not only on the current state but on the past history.

The reviewer's sentiment from the two statements above ("After all, the presented method is based, like several

300   previous pCO2 gap-filling studies, on instantaneous relationships to physical or biological oceanic quantities. While such relationships have proven to capture a good fraction of pCO2 variability, it is clear that oceanic biogeochemistry is a dynamical system, ie., pCO2, depends not only on the current state but on the past history") has been incorporated into the text on **L735**.

Though the need of "new methods" is being mentioned (L629-L638), the discussion remains solely with

305   regressions. For example, it ignores that other approaches like "data assimilation" into process models do exist already (though mostly not yet in a stage to fit the data closely). On the other hand, the discussion likewise ignores that sophisticated methods like regressions against drivers or data assimilation are only needed because large data gaps need to be bridged. In a data-rich world, such as pleaded for by the authors, simpler auto-regressive methods are also sufficient, as indicated e.g. in your Fig 10 by the relatively good agreement of

310   driver-regressions and auto-regressions in the more data-covered areas. In order to make the discussion more interesting in the revised paper, therefore, I feel that it should be done in a wider context of the existing literature and make more concrete statements on how to go forward.

The reviewer mentions, these methods are "not yet in a stage to fit the data closely". We have made the point that until we reach a stage when data assimilation methods are able to "fit the data closely", the community

315   should continue to explore machine learning alternatives. **L735:** *This includes assimilative modeling approaches, such as B-SOSE (Biogeochemical Southern Ocean State Estimate), which would also provide greater understanding of the driver for changes in surface pCO2 (Verdy and Mazloff, 2017). These methods may be able to provide better constraints on pCO2 in data poor regions. However, these assimilative models are not yet in a stage to fit the data closely (Verdy and Mazloff, 2017).*

320   An alternative option may be to substantially shorten the discussion and keep ideas for a future research paper (why not in the context of a new SOCOM as the authors propose?).

We would like to submit this manuscript as one article, rather than in two separate manuscripts.

The same also applies to the discussion of sampling strategies. The dependence of accuracy on data density, the need for denser sampling in many parts of the southern hemisphere, or the use of synthetic data to test sampling strategies, are all not new. While autonomous sampling devices are presented as "a new way", there are papers, e.g., from the SOCCOM project, which are not even cited in the discussion. These papers already discuss possibilities and limits on a higher level. In my opinion, a discussion of sampling also needs the input by the experimentalists (e.g., I'm not sure how "low-cost" the autonomous platforms really are). I feel the paper would win from a shorter and more concise discussion.

We have broadened the context of the discussion and simplified it as much as possible. We have also included references to ongoing projects, particularly the SOCCOM project and the breakthroughs that have been made in Southern Ocean $CO_2$ fluxes (Williams et al. 2017; Gray et al. 2018). We make the point these data have yet to be incorporated into regressive machine learning methods. We have removed the paragraphs dealing with scale-sensitive sampling as this is indeed a premature claim without sufficient backing of its efficacy to reduce the uncertainties in pCO2 estimates. These changes have been made from **L709 – L768** in the track changes document.

The authors find that the average over their ensemble of regressions performed better under several metrics than the individual ensemble members. They present this as a contradiction to warnings against the use of ensemble averages in the literature. When comparing the presented ensemble of pCO2 interpolations with e.g. the SOCOM ensemble, however, there seem to be distinct differences in how statistically homogeneous these ensembles are: The presented ensemble of regressions against the same explanatory variables likely spreads in the finer details (see the rather similar behaviour in Fig 6) such that averaging may reduce noise, while there are large systematic differences (including members with limited ability to fit the data) in the SOCOM ensemble or other ensembles from the literature. Therefore, I feel the authors should discuss better the conditions under which the average over an ensemble really is a meaningful estimate.

We removed the following quotes from SOCOM study:

"*We also discourage any ensemble averaging (or medians, etc.) of full spatiotemporal fields or time series, as this would result in variations that are not self-consistent any more and fit the data less well than individual products.*"

"*It also makes the case for ensembles stronger as the CSIRML6 performs well relative to other gap-filling methods.*"

The removal of these statements simplifies the subsection tremendously. IQR[14]/pco2 trend amplitude also offers a metric by which to scale data.

**Minor comments:**

L47: maybe add "e.g."

L47: done as suggested

L48: "maximise" does not seem to be the right word here

L50: Bold text added to make clearer: *The regression methods try to maximise **the utility** of existing ship-based observations **by** extrapolating $CO_2$ using proxy variables*

L57: "consolidated" probably means "collated" or similar

L59: Changed to collated

L65: as far as I see, there is not actually any weighting in that paper

L67: We have changed this to not focus on the weighting. The aim is to introduce the topic of RIAV, SOMFFN and MLS: *While SOCOM intercomparison did not **seek to** identify an optimal mapping method, it **assessed**  members according to…*

*Two methods, …, **achieved lower**  $R^{iav}$ scores **compared to other members of the comparison.***

L67: from my knowledge of the literature, most studies analysing existing pCO2 interpolations actually use several of these, rather than one "most widely used method"

L69: *The MPI-SOMFFN , is a global implementation of a two-step clustering-regression approach and has been  widely adopted  in the literature*

L106:maybe add "separately" after "applied"

L112: added *separately*

L107: "K-means clustering" should be briefly explained (either here or later, e.g. around L232)

We added/changed the following around **L228**: * We also use K-means clustering, **which groups data based on Euclidean distances. More** specifically, **we implement**  mini-batch K-means from  in Python's Scikit-Learn package*

L115: "described by" may better be "denoted as" (same in L120)

L120: Changed as suggested

L117: you probably mean "a range of 11 to 25 clusters"

L223: Changed as suggested

L123: spurious "and"

L128: Removed

Fig 1, section "DATA": Table "XX"

Replaced XX with 1.

Fig 1 Sect 4 (and many other places in text, tables, and figures): "HOTS" should be "HOT"

Corrected for all figures and tables

L140: maybe add "a" before "predictive"

L145: "a" added

L155: Explain or spell out "GHRSST"

L159: We have made a correction to the SST data citation, this is now OISST

L172: Is the use of "random noise" a standard technique to fill incomplete input data? Isn't there a chance that this creates instability to the regression? A brief explanation or a reference would be useful here

L188: *filled with low concentration random noise* **to be consistent with other regions of low concentration Chl-a (Gregor et al. 2017).**

L188: Maybe add "separate" or "individual" before "regressions"

L206: Added individual here

L198: "palate" is probably misspelled, what about "shown by separate colors"

L221: Changed as suggested

L214: Reference to Fig 5 would break the order of figures, but could easily be removed here

L234: Removed reference to Fig 5

L216: not sure "a.k.a." is a suitable abbreviation

L240: Removed the phrase in the brackets as *supervised learning* can also refer to classification.

L217: "80:20" seems to contradict "75:25" in Fig 1

Text is correct. Changed figure

L224-235: I found this paragraph difficult to understand. Can you say more explicitly which "hyper-parameters" you mean? It may also help to better link this paragraph and the previous one.

L248-L256: Linked the paragraph a little better, and referenced the supplementary materials where the model specific hyper-parameters are mentioned. Code has also been made available for the exact training procedure. Following text was removed and the paragraph was shortened:

~~Machine learning models have the ability to be as complex as the dataset at hand and are thus at risk of fitting not only the signal but also the noise of the training data – this is known as the bias-variance tradeoff. High variance is a result of a machine learning model that is too complex and is fitting the noise, and high bias is due to insufficient complexity where the model cannot fit the signal (Hastie et al. 2009). Machine learning algorithms have hyper-parameters that control the complexity of the model for each specific problem. In this study, hyper-parameters are tuned by training the model with grid-search cross-validation.~~ **We further reduce the possibility of overfitting by tuning the hyper-parameters for each model to be more generalised, i.e. able to fit the data that the model has not been exposed to. The search for the optimal hyper-parameters is achieved with grid-search cross-validation,** *where a portion of ...*

L244: add "below" after "Eqn 3 and 4"

L271: Added below

L247-257: I did not find this paragraph very clear. Does it mean that you repeat the previous steps with other selections of years in the test-training split?

L279: Changed the following sentence to make the paragraph clearer. Also referenced Figure 1 step 3 later in the paragraph as we feel that this illustrates the approach well.

*To overcome this limitation, we* ***iteratively*** *apply the train-test split method* ***with multiple selections of years***

L300: The RˆIAV metric was first mentioned in the SOCOM paper (ie. Rödenbeck et al., 2015)

L338: Corrected by removing the reference to Rödenbeck et al (2014)

L303-304: According to Rödenbeck et al. (2015), their benchmark has no interannual variability, but it does have seasonal variability.

*L342: ... by normalising annually weighted RMSE to a benchmark with  interannual  variability **driven only by atmospheric $pCO_2$:***

L306-307: Also here, the metric used by the authors (or at least the description of it) is not the same as that presented in Rödenbeck et al. (2015): SOCOM used the standard deviation over yearly averaged pCO2 mismatches, not standard deviations over the full data in individual years. If the metric used in this study has indeed been calculated in the way it is described, it should be sensitive to within-year variations (probably dominated by the match to the seasonal cycle) but be insensitive to interannual variations.

L345: AV was calculated correctly but described incorrectly. The text has been changed as follows: *where IAV has been removed by summing the climatology of the mapped $pCO_2$ and the annual trend of atmospheric $pCO_2$.*

L309-310: A benchmark constructed this way still contains the interannual variations of pCO2 (as the atmospheric pCO2 has very little IAV compared to seawater pCO2, except for the rising trend). Also here, this is opposite to what has been described by Rödenbeck et al. (2015) which removed IAV from the benchmark.

Addressed in the statement above (incorrectly described)

L313: I guess you mean "in contrast to the standard deviation which is sensitive to outliers."

L352: Changed as suggested

L314: It is not clear to me what "interannual resampling" means, please be more explicit.

L354: Changed to *annually averaged*

L319: What do you mean by "second part of the experiment"? Is it the "regression step of the algorithm"?

L359: Text changed as follows: *The results from the regression comparisons  (step two  in Figure 1)*

Fig 5: Why can e.g. D ever be worse than C, given that D has more degrees of freedom than C?

While there may be more degrees of freedom w.r.t. data clustering, the number of clusters (y-axis) is still kept constant. Moreover, the given clustering variables may not be suitable for separating the variability of pCO2. Hence, it is possible that regressions using the A clustering scheme (which has only 3 clustering variables), has slightly lower RMSE values compared to even D (which has 5 clustering variables).

L326-327: If Fig 5 is showing the averages of the scores from the 4 methods, I was wondering whether these 4 methods show the same general behaviour (ie. whether the better/worse scores occur in similar rows/columns)? I'm asking because only if yes, Fig 5 would give a meaningful picture about which number of clusters and which features are best. A statement on that should be added.

L373: We have included the regression-method-specific plots in the supplementary material. The distribution of these differs somewhat to Figure 5, but this is more transparent. We motivate that we use only the K21E clustering configuration for the sake of consistency. The following text was added on **L378-384**: *While the individual regression methods' bias and RMSE scores (Figures S5 and S6 respectively) do not match the distributions exactly, the two selected clustering configurations (black boxes in Figure 5) score consistently low for both metrics (with the exception of ERT – discussed in greater detail further on). We motivate to select only*

*one clustering configuration for the sake of simplicity. Furthermore, we select the configuration with 21 clusters (rather than 23) as fewer clusters further reduces the possible complexity at little cost..*

L343: This has been said before and should be omitted here.

L390: Removed second reference to BIO23

L344: Contradictory use of the term "ensemble", see remark above

Have changed *ensemble* to *ensemble average* where applicable

L347: add "average" after "ensemble"

Have changed *ensemble* to *ensemble average* where applicable

Tab 3 (also Tab4): I think the 1st column should better termed "clustering"

Table 3: Good suggestion. Done

L355: I guess you mean "for each number of clusters"?

L403: Changed to: *in each cluster**ing approach***

L366: I was wondering whether the occurrence of lower biases in the test years may actually be systematic (ie. not by chance). Are the same years (as listed in Sect 2.4) used in all the regressions? If so, couldn't it be that they implicitly lead to low biases through the model selection?

Yes! There is a chance that this may happen, and that is why we apply the robust approach in estimating RMSE as shown in Figure 6 and Table 4 where all years are equally represented.

L380: You probably mean "sampling density"?

L429: Added density

L397 and 399: The "||" around the unit is a rather sloppy notation. Better be explicit by writing "|bias| < 5uatm, RSME < 10uatm".

L448: Changed as suggested

L410: Duplicate "bias"

L460: Removed

L411: Add comma after "ERT", otherwise difficult to read.

L461: Added comma

L426: Are the statistics shown in the Taylor diagrams calculated over all individual data points? That is, do they reflect both spatial and temporal features?

L470: Yes, calculated over individual data points. Have added this to the text.

Fig 8: Consider to use the same radial axis limits for all 6 Taylor diagrams. For example, the estimates seem to lie more apart for HOT, but that's only because the variability at HOT is smaller than in others of the independent data sets.

X-axis limits changed. HOT(s) also renamed

L436: Spurious "that"

L489: Removed

L469: "2002" seems to contradict "2000" in L475 and 480.

L516: Corrected

L470: Shouldn't "regions" be "region groups"?

L517: No, regions refer to the grouped biomes.

L479: It seems to me that "reflect" would better fit than "highlight"

L531: Changed as suggested

L523-525: You could nicely link this back to Fig 2

L577: *The interannual estimates of interquartile range (IQR$^{IA}$; Figure 11a) show the disagreement between methods is relatively small in the majority of the ocean (≈ 5 µatm).* *he exceptions being the South Atlantic, southeastern Pacific and eastern equatorial Pacific with differences of > 10 µatm,* **where these regions coincide with regions of low sampling density (Figure 2)**.

L526: Add "trend" after "NH-ST", as it is mainly the trend which is reflected by IQRˆIA (if I understood correctly)

L579: Correct, added *trend*

L544: replace "ensembles" by "ensemble averages" (see remark on terminology above)

Done

L593, Fig 12: It remains unclear to me how to interpret the "seasonal cycle reproducibility". Doesn't it get smaller with stronger IAV? A short explanation would be helpful.

The image caption contains the description of what seasonal cycle reproducibility is

L611: duplicate "first"

L660: Removed

L628: According to the paper, the resolution is daily, not 6-hourly.

L677: Correct, changed to daily

L623: What do you mean by "procedural architectures"?

L732-L739: This paragraph has been reworded extensively and moved to the specified line numbers

L633: The method by Denvil-Sommer et al. (2018) is named as an example of a "fundamentally new method". In fact, however, the "CARBONES-NN" contribution to the SOCOM ensemble also employed a climatological and an interannual step, but did not outperform other methods there. To clarify this interesting question, I'd suggest to include the Denvil-Sommer et al. (2018) results into your comparison Sect 3.3-3.5, as this would allow a clean comparison.

We agree that Denvil-Sommer's method is not fundamentally new and this paragraph has been moved to the next section and simplified. The paragraph now includes reference to assimilation models (*e.g.* B-SOSE by Verdy and Mazloff 2017).

L645: If I understood correctly, the IQRˆIA metric is specifically sensitive to the trend. Why do you particularly propose this metric in the sampling context?

Have removed the reference to IQR^IA here due to extensive restructuring as suggested in the overall comments.

L695: Is "spatial coherence" really the right word here?

L788: Removed spatial, the rest of the sentence implies that we refer to the spatial distribution

[revised manuscript text omitted]

---

## Author Response (AR2)

**Response to R2**

We thank the reviewer for their detailed feedback and interest in our work! We hope that we have managed to correctly convey the reviewer's sentiment in the updated manuscript. Our responses are listed below in green. R2 is in black.

I thank the authors for responding in full to all of my questions and comments. However, in doing so they have introduced errors within their manuscript, and in some cases made incorrect statements and/or misunderstood concepts. These errors and misunderstandings need to be corrected before the manuscript is published. One error that the authors have made (in relation to uncertainties in the GLODAPv2 data) may change some results. So I recommend that the paper is revised before acceptance.

**Errors to be corrected**

1) The authors have attempted to address my questions about temperature handling, but they have misreported and/or misunderstood multiple aspects. These are:

   a) **lines 159 – 164.** These statements are misleading and they confuse the issue. The re-analysis is needed to ensure that all of the data are valid for a consistent depth. Ship and in situ measurements within a large collated dataset will have been collected from multiple different depths, and these variations in depth will have an unknown impact on the calculated air-sea gas fluxes. This problem is mentioned in Goddijn-Murphy et al, (2015) and explained in detail within Woolf et al., (2016). The use of a remote sensing temperature dataset for the re-analysis means that the data all become valid for a fixed and known depth. The Reynolds OISST dataset that the authors have used is a long term consistent climate quality dataset of remotely sensed infrared data (measurements of the thermal signal from the top few mm of the water) that are calibrated to fixed depth using quality-controlled and calibrated buoy in situ data. The statements on lines 159 to 164 (and the equivalent sentences in the supplementary section, and similar sentences in the discussion – see point 1e below) need to be updated so that they correctly convey this reasoning and explanation.

   We have rewritten this paragraph as the reviewer suggested. We have also removed the section in the supplementary materials that addressed this.

   The re-analysis to a known and consistent depth then also makes it possible for a more accurate gas flux calculation, as you then have the ability to use or estimate skin and sub-skin temperatures. But this is a separate issue.

   The product we use, OISSTv2 AVHRR only, reports bulk temperature - this puts it in an awkward place, where the corrections suggested in Woolf et al. (2016) recommend that skin or foundation temperatures are used. However, in the updated version, we aim to be consistent with the SOCOM study, so that we can compare our study with those independent studies in a meaningful manner. This means that we use the formulation for bulk fluxes where $k_w$ is scaled

to 16 cm/hr.

The version of the Reynolds OISST dataset also needs to be stated (as there are two different methodologies, one uses passive microwave, the other does not).

We have already stated the version of the OISST product (AVHRR only). This has also been updated in Table 1.

Woolf et al., (2016) https://agupubs.onlinelibrary.wiley.com/doi/full/10.1002/2015JC011427

Goddijn-Murphy et al, (2015) https://www.ocean-sci.net/11/519/2015/

b) **lines 165 to 167.** This is an unusual scientific justification. It reads like the authors are saying that they are choosing to exclude recent advances, as including them appears to degrade their expected result. It's also not clear if the 'reference' dataset (that they used for the evaluation in the supplementary section) was the original SOCAT data or the re-analysed SOCAT data. You would expect that assessing a re-analysed (interpolated output) result, using the original (not re-analysed) dataset as the reference would indeed give a poorer result (due to the poorer reference value).

We have removed this section from the supplementary materials.

This latter point is an example of why using RMSE, where the E is error, is potentially misleading as it's a difference, not an error – please see my point 3 below.

A better reasoning for their method choice would be that the authors are choosing a method that is consistent with those used by the SOCOM datasets, to which the authors would like to compare/contrast their work.

We have changed our reasoning as suggested by the reviewer. L379-181 now reads: *While this is an important consideration, we do not apply a temperature correction as we aim to be consistent with the pCO$_2$ estimates from the SOCOM intercomparison (Rödenbeck et al., 2015).*

c) The methods are confusing and inconsistent with the response to reviewer comments. In the reviewer responses the authors say that they will use a more accurate calculation for the gas fluxes (as given by Woolf et al., 2016 where two solubilities are used), whereas equation 2 says that they are not using a more accurate calculation (as it contains one solubility term), but then on line 315 the authors say that they are using a more accurate calculation (they state they are using the Rapid model, but Equation 2 is not the rapid model). Please, can the authors clarify their methods and ensure that the text matches the equations and the information from the Woolf et al, (2016) reference that they refer to.

We have chosen to use the bulk flux approximation scaled to 16 cm/hr after Rödenbeck et al. (2015). Scaling the fluxes is a recognition that there is uncertainty in the bulk flux approximation. This section (L345-352) now reads: *One of the problems with the bulk estimates of sea-are CO$_2$ fluxes is that models of gas exchange in the surface layer of the water column are simplified, but there are approaches, such as the rapid equilibrium model, that account for more complex temperature gradients in the upper layer of the surface ocean*

*(Wanninkhof et al. 2009; Woolf et al. 2016). However, for the sake of consistency with past studies, we use the bulk approximation of sea-air fluxes (Eq. 2) where $k_w$ is scaled to 16 cm.hr$^{-1}$ as in the SOCOM intercomparison (Rödenbeck et al., 2015).*

d) Why is the UEA-SI method an exception? (as stated on line 454 to 455). All of the SOCOM datasets that the authors have used the original SOCAT data and so none of these datasets have re-analysed the SOCAT data. So I don't think that there are any exceptions and they will also contain unknown impacts due to not re-analysing the SOCAT data to a consistent depth.

The text *with the exception of UEA-SI* has been removed

e) **lines 655 to 664.** This paragraph is also confused and makes incorrect statements. The same mistakes as described above in point 1a) are again repeated in this paragraph. please refer to the original Woolf et al., 2016 paper where these issues are discussed in depth.

Any difference introduced due to inconsistent sampling depths will vary spatially and dependent upon the ship collecting these measurements, and so the measurements (original versus re-analysed) are unlikely to always shift in unison as suggested. Eg. there are likely to be larger differences between original and re-analysed data within an area of upwelling, in comparison to areas of well-mixed ocean waters.

The paragraph now reads (L736-740): *Another source of bias is the mismatch between the temperature at which $pCO_2$ is measured (i.e. at the depth of a ship's intake) and the temperature to which $pCO_2$ is predicted (~1 m in the case of the dOISSTv2 data; Banzon et al. 2016; Goddijn-Murphy et al. 2015). Goddijn-Murphy et al. (2015) show that this mismatch is considerable in some cases. However, this correction also makes the assumption that temperature is the only factor that influences $pCO_2$ in the surface layer of the ocean. The correction will thus not account for other processes such as primary production, stratification and gas exchange within the surface layer. This is an issue that should be discussed by the community and tested empirically to assess the impact that processes other than temperature difference may have on $pCO_2$.*

2) The authors have used an incorrect combined uncertainty value for the GLODAPv2 dataset. The value taken from the Olsen et al., (2016) paper that the authors have used as a combined uncertainty is actually the bias value, which is only one part of a Type A uncertainty (BIPM, 2008). Both a bias and variance estimate are needed for a Type A uncertainty estimate. The Olsen et al. work does not provide a variance estimate, so I would suggest that the authors use the state of the art uncertainties as provided by Bockmon and Dickson, (2015). Bockmon and Dickson give a combined uncertainty of 0.5% for total alkalinity and for total dissolved carbon (whereas the existing bias estimate that the authors are using equates to ~0.2% for alkalinity, which illustrates that the uncertainty estimate used by the authors is too low). Using a more correct estimate of the combined uncertainty may mean that some results throughout the paper need updating.

We thank the reviewer for highlighting this. The following sentence has been added on line 323:
*However, this error may be larger as reported in Table 2, where Bockmon and Dickson (2015) showed*

*that the uncertainty for DIC and TA is likely closer to ±10 µmol.kg⁻¹.* This error is consistent in the comparisons with the SOCOM products, thus is inconsequential in the context of this study.

   a) BIPM, 2008 https://www.bipm.org/utils/common/documents/jcgm/JCGM_100_2008_E.pdf
   b) Bockmon and Dickson, (2015)
      https://www.sciencedirect.com/science/article/pii/S0304420315000213
   c) Olsen et al., 2016 https://www.earth-syst-sci-data.net/8/297/2016/

3) Throughout the manuscript, the authors have incorrectly used the word 'error' when they actually mean uncertainty or difference (e.g. when using the root mean squared error, RMSE). The use of the word error implies that some 'truth' value is known, whereas all measurements and observations contain uncertainties and are not 'truth' (including those within GLODAPv2 and SOCAT that are being used as a reference for the RMSE calculations). I would suggest that the authors instead report RMSD (where the D is difference) and check and correct the wording used within sentences throughout the paper (and the supplementary) to refer to differences, uncertainties and errors (as many instances where the word 'error' has been used are not strictly 'errors').

I have tried to find a reference that substantiates the reviewers terminology, but I could not find any. RMSE is also a widely used term in machine learning literature. We have thus chosen to stick with the RMSE terminology. We have also clarified our definition of "error" with: "Note that the term "error" refers here to the error introduced by the gap-filling method relative to the observations."

**Typographical errors**

There typographical errors in the main text. (e.g. 'sea-are' on line 311) and in the reference list (e.g. the Woolf et al., 2015 paper in the reference list on page 35 has incorrect lead author, incorrect co-author list and incorrect publication year). There are also sentences that are incomplete and/or contain errors (e.g. line 123 in the supplementary '…showing that SOCAT temperatures or on average… [whereas it should be 'temperatures are on average']).

I would encourage the authors to take some time and slowly go through the manuscript and check their sentences, phrasing, spelling and references.

We apologise for the multiple typographical errors and the incorrect listing of the reference – this has been removed as fluxes are calculated using bulk estimates in the final form. These corrections and others have been made throughout the document.

[revised manuscript text omitted]

---

## Author Response (AR3)

**Revision to editor's comments:**

Dear Prof Yool,

We thank you for your last comment.

I have uploaded our data to the OCADS server. This has been updated in the **Code and data availability** section:

> *Supporting code is available in Supplementary Materials. Data (global surface ocean $pCO_2$ and $FCO_2$ from CSIR-ML6 version 2019a) is available at Ocean Carbon Data System (OCADS, [https://www.nodc.noaa.gov/ocads/oceans/ndp_101/ndp101.html](https://www.nodc.noaa.gov/ocads/oceans/ndp_101/ndp101.html)).*

The metadata of for this data is available and Alex Kozyr (data manager) said that the data should be downloadable soon with a DOI. I'm sorry this has taken so long, but this is unfortunately out of my control at the moment.

I hope that we can move forward with this information.

Regards

Luke (on behalf of the authors)